# A Narrative Review of Prognostic Gene Signatures in Oral Squamous Cell Carcinoma Using LASSO Cox Regression

**DOI:** 10.3390/biomedicines13010134

**Published:** 2025-01-08

**Authors:** Nur Fatinazwa Mohd Faizal, Saptarsi Shai, Bansi P. Savaliya, Lee Peng Karen-Ng, Rupa Kumari, Rahul Kumar, Vui King Vincent-Chong

**Affiliations:** 1Oral Cancer Research & Coordinating Centre (OCRCC), Faculty of Dentistry, Universiti Malaya, Kuala Lumpur 50603, Malaysia; ftnzwatwork@gmail.com (N.F.M.F.); karennlp@um.edu.my (L.P.K.-N.); 2Baylor College of Medicine, Texas Children’s Hospital, Houston, TX 77030, USA; saptarsi.shai@bcm.edu; 3Division of Nephrology and Hypertension, Mayo Clinic, Rochester, MN 55901, USA; bansisavaliya33@gmail.com; 4Department of Pharmacology and Therapeutics, Roswell Park Comprehensive Cancer Center, Buffalo, NY 14263, USA; rupa.kumari@roswellpark.org; 5Center for Oral Oncology, Roswell Park Comprehensive Cancer Center, Buffalo, NY 14263, USA

**Keywords:** OSCC, HNSCC, LASSO, chemotherapy, immunotherapy, prognosis, gene signature

## Abstract

Oral squamous cell carcinoma (OSCC) is one of the most common malignancies of the head and neck squamous cell carcinoma (HNSCC). HNSCC is recognized as the eighth most commonly occurring cancer globally in men. It is essential to distinguish between cancers arising in the head and neck regions due to significant differences in their etiologies, treatment approaches, and prognoses. As the Cancer Genome Atlas (TCGA) dataset is available in HNSCC, the survival analysis prognosis of OSCC patients based on the TCGA dataset for discovering gene expression-based prognostic biomarkers is limited. To address this paucity, we aimed to provide comprehensive evidence by recruiting studies that have reported new biomarkers/signatures to establish a prognostic model to predict the survival of OSCC patients. Using PubMed search, we have identified 34 studies that have been using the least absolute shrinkage and selection operator (LASSO)-based Cox regression analyses to establish signature prognosis that related to different pathways in OSCC from the past 4 years. Our review was focused on summarizing these signatures and implications for targeted therapy using FDA-approved drugs. Furthermore, we conducted an analysis of the LASSO Cox regression gene signatures. Our findings revealed 13 studies that correlated a greater number of regulatory T cells (Tregs) cells in protective gene signatures with increased recurrence-free and overall survival rates. Conversely, two studies displayed an opposing trend in cases of OSCC. We will also explore how the dysregulation of these signatures impacts immune status, promoting tumor immune evasion or, conversely, enhancing immune surveillance. Overall, this review will provide new insight for future anti-cancer therapies based on the potential gene that is associated with poor prognosis in OSCC.

## 1. Introduction

Head and neck squamous cell carcinoma, known as HNSCC, is ranked as the seventh most common malignancy globally and is a significant public health concern due to its high rates of morbidity and mortality [1,2]. According to estimates from Global Cancer Observatory, HNSCC is responsible for approximately 890,000 new cases and 450,000 deaths each year, representing approximately 4.5% of all cancer diagnoses and deaths [1,2]. Current treatment regimens for HNSCC focus on surgery, chemoradiotherapy, targeted therapy and immunotherapy and are yet to improve the survival rate of HNSCC in recent decades [3,4,5,6]. Moreover, these treatments also lead to complications resulting from the death of non-characteristic cells [4]. Therefore, it is imperative to explore potential prognostic indicators and treatment targets for HNSCC to properly control the treatment intensity and prevent serious complications brought on by overtreatment. 

### 1.1. Clinical and Molecular Features of OSCC

Oral squamous cell carcinoma (OSCC) is a common type of HNSCC and is recognized as the eighth most common cancer among men worldwide [7] accounts for over half of these cases and originates in the squamous epithelium of the oral cavity or lip, including various oral structures such as the labial mucosa, buccal mucosa, floor of the mouth, and anterior two-thirds of the tongue [8,9]. The oropharynx, which includes the soft palate, base of the tongue, palatine tonsils, and posterior pharyngeal wall, is structurally distinct from the oral cavity, separated by the junction of the hard and soft palate above and the circumvallate papillae below [9]. Regrettably, 60% of OSCC cases are diagnosed at advanced stages (III and IV), resulting in a dismal prognosis with a 5-year survival rate of merely 30% [7,10,11,12]. Major risk factors for developing OSCC are excessive alcohol consumption, tobacco use, betel quid chewing, and infection with human papillomavirus (HPV) [2,13]. In fact, HPV-related HNSCC has a higher incidence in the oropharynx, hypopharynx, and larynx compared to the oral cavity and is linked to a notably longer median survival period of 130 months as opposed to 20 months [14]. These findings indicate significant molecular distinctions between HPV-positive HNSCC, predominantly originating from oropharyngeal tumors, and HPV-negative HNSCC, which is often associated with heavy tobacco/betel quid consumption and is more frequently detected in the oral cavity.

### 1.2. Genomic Alterations and Molecular Subtypes of OSCC

In terms of their genomic alterations, HPV-negative HNSCC displays a high frequency of mutations in *TP53* (83%) and *CDKN2A* (57%), as per data from The Cancer Genome Atlas (TCGA) of HNSCC [15]. Conversely, HPV-positive oropharyngeal cancer often exhibits *PIK3CA* amplifications/mutations (56%), with other genetic changes being less common. Additionally, OSCC, particularly in gingivo-buccal SCC prevalent in regions where tobacco-chewing is widespread, is characterized by mutations in specific cancer-related genes such as *USP9X*, *MLL4*, *ARID2*, *UNCBC*, and *TRPM3* [16]. Meanwhile, genes like *TP53*, *FAT1*, *CASP8*, *HRAS*, and *NOTCH1* are commonly mutated in all HNSCC cases [15,16]. This distinctive mutational profile identified predominantly in gingivo-buccal SCC underscores the importance of tumor suppressor genes over oncogenes in the development of OSCC. The prevalence of *TP53* mutations and *p16* deletion in non-HPV-related HNSCC, as indicated by analysis of TCGA data, further supports this finding.

In general, OSCC exhibits a higher level of tumor heterogeneity compared to other types of carcinomas in the body, which has implications for the development of targeted therapies [17,18]. For instance, it has been observed that tongue SCC tends to exhibit more rapid growth than other types of oral cavity cancers, with a higher incidence of cervical metastases. In terms of the molecular landscape, Severino et al. [18] have illustrated a distinct molecular landscape disparity between tongue and floor-of-mouth SCC, underscoring the molecular diversity present within specific regions of the oral cavity. This finding is further corroborated by the research of Chung et al. [19], whose study strived to categorize tumors into more homogeneous groups based on their gene expression patterns, a task made challenging by the heterogeneous nature of OSCC as compared to those originating from the oropharynx, hypopharynx, and larynx. Eliminating OSCC from the analysis significantly improved the predictive accuracy of prognosis in their study, highlighting the unique and heterogeneous characteristics of OSCC as compared to non-OSCC. Moreover, Sathyan et al. [20] investigated potential heterogeneity in the cell cycle regulatory mechanisms of buccal mucosal and tongue OSCC by comparing key cell cycle regulatory proteins, finding significant downregulation of *p16* and *p21* in tongue SCC compared to buccal SCC. Additionally, research by Kannan et al. [21] examined telomerase expression in tongue and buccal mucosa SCCs, revealing that a majority of tongue SCCs exhibited weak or negative telomerase activity across all stages, while most buccal SCC showcased varying levels of telomerase activity. Furthermore, Vincent-Chong et al. [22] have demonstrated contrasting genetic profiles and biological pathways between SCCs originating in the tongue and cheek. 

When analyzing the molecular subtypes derived from atypical, basal, classical, and mesenchymal of both OSCC and non-OSCC, Foy et al. [23] conducted a study using data from the TCGA HNSCC public repository. They identified OSCC portrayed basal and mesenchymal as the main subtypes, while atypical and classical were predominant in non-OSCC tumors, such as those found in the larynx, oropharynx, and hypopharynx. In a separate study by Zhang et al. [24], molecular landscape information from the TCGA HNSCC cohort was used to assess the molecular characteristics of these subtypes (atypical, basal, classical, and mesenchymal) to categorize for immune response and prognosis. They identified Cluster 2, which mainly consisted of basal and mesenchymal subtypes that were more common in OSCC. These subtypes showed higher levels of intermediate immune infiltration and cold immune features, as well as lower levels of hot immune features and activated immune responses, compared to Cluster 1, which predominantly included atypical and classical subtypes. When examining survival rates, patients in Cluster 1 displayed improved overall survival (OS) and progression-free survival (PFS) rates. It was also observed that Cluster 1 exhibited more inhibited oncogenic signaling, greater immune cell infiltration, increased responsiveness to immunotherapy and chemotherapy, and the most favorable prognosis.

### 1.3. The Immune Landscape of OSCC

In terms of the immune landscape, OSCC exhibits a unique immune profile when compared to other HNSCCs, such as those in the oropharynx, larynx, and hypopharynx [25]. Oropharyngeal SCC, for example, demonstrates higher levels of T cell infiltration and immune activation [12]. Conversely, the OSCC shows the lowest CD8/regulatory T cells (Tregs) ratio among these subsites, suggesting a less robust immune response and potentially weaker anti-tumor immunity. This discrepancy in ratios within the oral cavity implies a more immune-suppressive environment, which could lead to compromised immune surveillance and reduced responsiveness to immunotherapy in comparison to other HNSCC subsites [26]. A study by Muijlwijk et al. [27] has highlighted variations in the tumor immune microenvironment (TIME) among HNSCC subsites, with OSCC exhibiting the highest T cell infiltration when contrasted with HPV-negative SCC in the oropharynx, larynx, and hypopharynx. However, this heightened infiltration in OSCC is predominantly characterized by elevated levels of CD4 T helper (Th) cells and a larger proportion of Tregs rather than CD8 cytotoxic T cells. Despite the increased presence of immune cells within OSCC tissue, adjacent non-tumor tissue does not exhibit similar levels of immune infiltration. These findings suggest that, despite the overall increase in T cell presence in OSCC, the immune response may be more regulated or suppressed due to the abundance of Tregs. This disparity in the balance between Th cells and Tregs indicates a potentially more immune-suppressive microenvironment in OSCC compared to other HNSCC subsites, which could impact immune surveillance and limit the effectiveness of immunotherapies like checkpoint inhibitors. Conversely, subsites such as the oropharynx, particularly in cases of HPV-positive SCC, tend to display a more favorable CD8/Treg ratio, making them potentially more responsive to immunotherapy. Additionally, research by Spector et al. [12] has shown that oropharyngeal SCC has a higher number of tumor-infiltrating lymphocytes (TILs) due to the immunogenic nature of HPV-related tumors in the lymphoid-rich environment of the oropharynx, as opposed to OSCC. A study by Mandal et al. [26] also supports this trend, with oropharynx SCC showing higher levels of T cell infiltration, immune activation, and immunoregulatory influence characterized by a lower CD8/Treg ratio compared to other HNSCC, including OSCC. 

### 1.4. Significances of OSCC Gene Signatures

Given the distinct immunological and molecular characteristics of OSCC, it is essential to create gene signatures specific to OSCC for prognostic purposes, as they have the potential to reveal unique pathways impacting patient outcomes. The heterogeneous nature of OSCC, particularly its increased presence of Tregs and CD4 T cells, underscores the differences in its immunological and molecular makeup compared to other areas of HNSCC, such as the oropharynx and larynx [27]. These variances in immunity and molecular profile are likely to influence tumor progression, response to therapy, and prognosis, thus making an OSCC-specific gene signature better suited to capturing the complexities of this tumor’s biology. Furthermore, gene signatures originating from HNSCC may highlight common pathways but could overlook important genetic mutations or immune responses that are specific to OSCC, like the frequent mutations in *TP53*, *FAT1*, and *CASP8* that are uniquely found in the OSCC [16]. By focusing on OSCC-specific gene signatures, we can pinpoint more precise prognostic indicators and treatment targets that are tailored to the distinct molecular drivers of OSCC. This targeted approach will ultimately enable more accurate patient classification and the development of personalized treatment strategies that address the specific challenges presented by OSCC.

To date, incisional biopsy remains the gold standard for OSCC diagnosis, along with therapeutic schedule and prognostic predictions based on TNM staging [28,29]. However, studies have indicated that TNM staging failed to meet the requirements for selecting treatment options and predicting prognosis [29]. Lately, regression analysis has been widely used in the study of cancer prognosis, including univariate Cox regression analysis, multivariate Cox regression analysis and least absolute shrinkage and selection operator (LASSO) regression analysis [30,31,32,33]. Data for these studies were downloaded from TCGA, which is a comprehensive open-access cancer genomics platform covering 30 different cancer types and over 11,000 specimens subjected to genomic expression, sequences, methylation and copy number. The successful utilization of LASSO in these scenarios emphasizes its adaptability and efficiency as a feature selection tool in the field of oncology. It not only enhances the accuracy of predicting gene signatures but also helps in unveiling the fundamental mechanisms steering cancer behavior. This strategy lays the groundwork for creating targeted treatments and individualized therapeutic approaches, ultimately enhancing outcomes for patients affected by a variety of solid tumors.

Between 2015 to 2020, few studies [34,35,36,37,38,39] reported that gene signatures and prognostic models based on regression analysis were constructed to improve clinical prognosis and direct careful design of better treatment plans for OSCC. However, these studies did not investigate the correlation of the gene signatures with immune status in OSCC patients. To date, no review paper has solely focused on providing a systemic overview of gene signature that is related to prognostic risk prediction models in OSCC that integrate with cancer pathways and immune status in OSCC. To address this gap in knowledge, we have provided a comprehensive review that covers the current prognostic risk prediction model of OSCC based on different signatures related to cancer pathways and immune status for the past 4 years. Apart from that, we will also highlight the future direction of how to utilize the cancer signature to predict clinical prognosis and assist in the formulation of OSCC treatment plans. Moreover, we will also provide the interaction of immune-related gene signatures that are associated with OSCC prognosis linked to the immune response/status. For the future direction, we are providing the implication for targeted therapy using FDA approved drugs for the gene signature using https://clue.io/repurposing-app (accessed on 1 June 2024).

### 1.5. Research Strategy

PubMed databases were searched from 2021 to 2024 using various combinations of the following keywords: “oral squamous cell carcinoma (OSCC)” and “least absolute shrinkage and selection operator (LASSO) regression analysis in oral squamous cell carcinoma (OSCC)”. Original experimental studies (both in vitro and in vivo), reviews, editorial letters, book chapters, opinions, and abstracts from the analyses published in English considering stem cell markers in OSCC were considered and included.

## 2. Prognostic Gene Signatures Related to Cancer Pathways in OSCC Using LASSO Cox Regression Analyses

In the oncology field, the identification of reliable gene signatures for predicting patient prognosis is increasingly vital for tailored cancer therapy [40]. A brief explanation of the pros and cons used in survival prediction in cancer research was illustrated in Appendix A. A rigorous method for creating prognostic models in solid tumors is the LASSO [41]. The justification for using LASSO Cox Regression in cancer research was illustrated in Appendix A. LASSO, a robust statistical technique for feature selection and regularization, is particularly valuable in handling high-dimensional datasets commonly found in the cancer genomics [42,43]. By penalizing and shrinking regression coefficients toward zero, LASSO effectively eliminates irrelevant variables and identifies only the most critical features for model creation. In prognostic research, LASSO has demonstrated high accuracy in developing gene signatures that can forecast patient outcomes with precision. Through the application of a shrinkage procedure to gene expression data, LASSO simplifies the model by retaining only the most important gene variables with non-zero coefficients. This process improves the interpretability and strength of the model, making it well-suited for studies on solid tumors where numerous genes may impact cancer progression and treatment resistance [44].

In this review, we have summarized a total of 34 studies that have employed LASSO Cox regression to generate the independent prognostic gene signatures related to OSCC, as depicted in Table 1, Table 2, Table 3, Table 4 and Table 5. Relevant studies showed that various cancer pathways provide information on patient prognosis and the treatment of cancer, including five main pathways with different signatures such as programmed cell death, epigenetic and gene regulation, immune response and tumor microenvironment (TME), metabolism and energy and resistance to cell death.

## 3. Programmed Cell Death

Programmed cell death (PCD) is a regulated and deliberate process through which cells intentionally end their own life in response to either physiological or pathological signals. In contrast to unplanned cell death resulting from physical or chemical damage, PCD is an active mechanism that is gene-regulated and essential for sustaining cellular balance, development, and defense against illnesses such as cancer. Traditional categories of PCD include apoptosis, which features caspase activation and chromatin condensation, and necrosis, a passive form of cell death that is caspase-independent and involves cell swelling and membrane breakdown. Recently identified types of PCD have broadened this classification, revealing various molecular mechanisms and their significance in health and disease. These encompass ferroptosis (which involves iron-dependent lipid peroxidation), autophagy (the process of self-digestion of cellular components), necroptosis (an inflammatory and caspase-independent form of death regulated by *RIPK3* and *MLKL*), cuproptosis (damage caused by mitochondrial toxicity due to copper), pyroptosis (inflammatory cell death mediated by gasdermin), and disulfidptosis (collapse of cytoskeletal proteins triggered by disulfide). Each subtype possesses distinct genetic, biochemical, and morphological features, and their irregularities are frequently associated with cancer development and therapy resistance. Gaining a deeper understanding of these PCD pathways sheds light on cancer biology and presents potential therapeutic avenues.

### 3.1. Ferroptosis

Ferroptosis is a recently found form of cell death that is frequently characterized by a huge amount of iron accumulation and peroxidation of lipids, which is genetically, morphologically, and biochemically distinct from apoptosis, necrosis, and autophagy [45,46,47]. In the recent past, ferroptosis has become a quiescent target for cancer therapy, particularly in cancers that are resistant to drugs and radiation [45,48,49]. Previous findings have also reported the role of ferroptosis in the OSCC [50,51,52]. Several genes, such as *SreBP* and *GPX4*, that aid the proliferation of OSCC cells were reported to protect cells from ferroptosis. Additionally, Zhu et al. [53] found that OSCC overexpressed the ferroptosis-negative regulatory gene *SLC7A11*. OSCC cells-induced ferroptosis was used to overcome hypoxia-related photodynamic therapy resistance. Various drugs, such as telaglenastat (CB-839) and quisinostat, as well as new materials like zero-valent iron nanoparticles, have been demonstrated to enhance anti-tumorigenesis response partly by promoting ferroptosis in OSCC [50,51,52].

A study by Li et al. [54] explored the predictive value of ferroptosis-related genes (FRGs) on the OS rate of OSCC. This study obtained the mRNA expression profiles of FRGs and clinical information of OSCC patients from the TCGA database. A total of 86 differentially expressed ferroptosis-related genes (DE-FRGs) were determined in both OSCC and adjacent normal tissues, of which 50 genes were upregulated and 36 genes were downregulated. The results of gene ontology (GO) and Kyoto Encyclopedia of Genes and Genomes (KEGG) analysis suggested that the DE-FRGs were significantly linked with ferroptosis rather than another form of cell death. Ten OSCC-prognostic FRGs were identified via LASSO regression analysis (*ATG5*, *BID*, *ACO1*, *GOT1*, *AKR1C3*, *GLS2*, ALOX15, *SCO2*, *MAP1LC3A*, *MAP3K5*). Among these FRGs, seven genes (*ATG5*, *BID*, *ACO1*, *GOT1*, *AKR1C3*, *GLS2*, *ALOX15*) were regarded as risk genes, and the other three genes (*SCO2*, *MAP1LC3A*, *MAP3K5*) were considered protective genes. In addition, the protein-protein interaction (PPI) network demonstrated that *ATG5*, *MAP3K5*, and *MAP1LC3A* were regarded as core genes. Then, patients were categorized into low- and high-risk groups according to the risk score cutoff. Patients in the high-risk group were reported to have a significantly worse OS. Based on single sample gene set enrichment analysis (ssGSEA) analysis, the high-risk group was highly associated with cell cycle, p53 signaling pathway, DNA replication, RNA degradation, and mismatch repair. Whereas the low-risk group is enriched with immune-related signaling pathways such as T cell receptor signaling pathway, B cell receptor signaling pathway, FcγR mediated phagocytosis, and primary immunodeficiency. This suggests that the 10-FRS signature may have a significant impact on immune-related, cancer-related signaling pathways. Dysregulation of these pathways could be linked to the progression and development of tumors. Immature dendritic cell (iDC), activated DC (aDCs), mast cells, plasmacytoid DC (pDC), type II interferon gamma (IFN) response, T cell co-stimulation, and B cells were significantly activated in the low-risk group. Meanwhile, immune effector cells such as CD8 T, TILs, natural killer (NK) cells, Th cells, and Treg were downregulated in the high-risk groups.

Fan et al. [55] conducted thorough research that revealed nine genes—*CISD2*, *DDIT4*, *CA9*, *ALOX15*, *ATG5*, *BECN1*, *BNIP3*, *PRDX5*, and *MAP1LC3A*—that are differentially expressed ferroptosis-related with prognostic significance in OSCC. Although *MAP1LC3A* was linked to enhanced survival outcomes, demonstrating its protective effect, Kaplan Meier analysis showed that lower expression levels of *CISD2*, *DDIT4*, *CA9*, *ALOX15*, *ATG5*, *BECN1*, *BNIP3*, and *PRDX5* were associated with better OS. Advanced stages (III–IV) were linked to *CISD2*, *ATG5*, and *DDIT4*, while elevated levels of *MAP1LC3A*, *PRDX6*, *CISD2*, and *ATG2* were tied with higher tumor grades. Low numbers of various immune cell types, such as B cells, CD8 T cells, DCs, and macrophages, as well as decreased immunological and stromal scores, were associated with high-risk scores. Additionally, these PR-DE-FRGs showed variable sensitivity to chemotherapy: *MAP1LC3A* correlated adversely with methotrexate, *BECN1* negatively with docetaxel, *ALOX15* positively with doxorubicin and etoposide, and BNIP3 negatively with cisplatin. Numerous TP53-related, immune-related, and ferroptosis-related activities and pathways were found to be significantly enriched in the various risk groups’ enrichment analyses. Further investigations of the immune microenvironment and mutations revealed associations between risk scores and immunity, as well as *TP53* mutations.

### 3.2. Autophagy

Autophagy is a highly conserved self-digesting process which often employed at basal levels but can be triggered in certain states of cellular stress, such as cancer, resulting in metabolic adaptation and nutrient cycling by transferring damaged cellular components to lysosomes for degradation [56]. Autophagy has a dual function in carcinogenesis; for instance, it can inhibit carcinogenesis in the early stage by eliminating damaged organelles and proteins to reduce cellular damage and maintain metabolic stability [57]. However, once cancer occurs, autophagy increases tumor cell survival in stressful conditions, thus promoting tumor growth [58]. Numerous studies have demonstrated the correlation between autophagy and OSCC. For example, a recent study suggested that autophagy played a role in maintaining stemness and promoting drug tolerance in OSCC [59].

Hou et al. [35] identified thirteen gene signatures based on autophagy-related genes (ARGs) that related to prognosis in OSCC and successfully divided OSCC patients into low and high-risk groups with significantly different OS. A total of 222 ARGs were obtained from the Human Autophagy Database (HADb). TCGA and Gene Expression Omnibus (GEO) databases were used to obtain RNA-sequencing (RNA-seq) and clinical information of OSCC patients. Cox proportional hazard regression analysis was used to screen the prognosis-related ARGs, and the acquired genes were subjected to LASSO Cox regression, and risk scores were calculated. Among the thirteen genes, four genes (*USP10*, *ATF6*, *MAPK9*, *BID*) were considered as risk genes, while the other genes (*FOS*, *MAP1LC3A*, *SPHK1*, *GRID1*, *IKBKB*, *RAB24*, *CFLAR*, *WDR45*, *RAF1*) were considered as protective genes. Patients were then separated into low-and high-risk groups with significantly different OS. These prognostic genes were subjected to GO, KEGG, ssGSEA analysis, and PPI networks to study the associated biological characteristics and signaling pathways. Lastly, to improve the precision of OSCC patients’ survival predictions, both gene signatures and numerous clinical parameters were integrated to create a robust prognostic nomogram. This finding aligns with previous studies that reported a high correlation of the MAP1LC3A gene with tumorigenesis in many cancers, including esophageal squamous carcinoma, gastric cancer, and osteosarcoma [60]. RAB24 and WDR45 genes were also associated with tumorigenesis [61,62]. This study suggests that 13-ARGs could hold promise as novel prognostic biomarkers and potential targets for therapy. Therefore, aiding clinicians in developing customized and optimized treatment strategies.

### 3.3. Necroptosis

Necroptosis is a type of programmed, caspase-independent, controlled cell death that follows necrosis. It is primarily dependent on receptor-interacting serine-threonine kinase 3 (RIPK3) and mixed lineage kinase domain-like (MLKL) and typically presents necrosis-like structural characteristics [63,64]. This process disrupts the cell membranes, leading to inflammation and vascularization. Interestingly, necroptosis has been shown to not only trigger inflammatory reactions but also stimulate immunosuppression and tumorigenesis [65]. Necroptosis has been found to have two opposing impacts on cancer: it can both promote and inhibit tumor growth, which is why it has been linked to both positive and negative outcomes in various cancer types [66,67]. While necroptosis’s roles in other tumor types have been established, little is known about how OSCC is treated by using the necroptotic process [67,68].

Six genes were found by Huang et al. [67] to be associated with necroptosis in OSCC: *HPRT1*, *PGAM5*, *BID*, *SMN1*, *FADD*, and *KIAA1191*. Among these, *KIAA1191* did not significantly differ from normal tissues in terms of expression, whereas *HPRT1*, *PGAM5*, *BID*, *SMN1*, and *FADD* did. There are two major risk genes identified: *BID* and *HPRT1*. Positive correlations were found between *HPRT1* expression and Th2 cells, Th cells, and Tgd cells. T cells, T central memory (Tcm), T effector memory (Tem), T follicular helper (Tfh), Th1 cells, Th17 cells, Treg, macrophages, mast cells, neutrophils, NK CD56 bright and dim cells, and T cells were found to have negative correlations with *HPRT1* expression. On the other hand, negative correlations were found with the expression of *HPRT1*. Similarly, *BID* expression was found to be negatively correlated with NK CD56 bright cells, NK cells, Th2 cells, and macrophages but positively correlated with DC, aDC, B cells, CD8 T cells, cytotoxic cells, eosinophils, iDC, mast cells, neutrophils, and various T cell subsets, including Th, Th1, Th17, and Treg. Given that functional investigations reveal that *HPRT1* knockdown in OSCC preclinical models reduces cell migration and proliferation, the protein may be involved in tumor progression. Emphasizing the complex relationships between necroptosis and the immune system in OSCC offers insights into potential therapeutic options for the treatment of this type of cancer. Necroptosis-related genes (NRGs) significantly influence the pathogenesis and prognosis of OSCC. A few key pathways produce necroptosis in OSCC. Initially, TNFα binds to the TNFR receptor on the cell membrane, initiating the production of complex I, complex II, and finally, the necrosome. This process causes MLKL to be phosphorylated, which in turn activates DRP1 and PGAM5, leading to mitochondrial fission necroptosis and cell death [69,70]. Moreover, the pro-apoptotic gene *BID* has two functions: it promotes both cell death and tumor growth by elevating cell proliferation. Throughout the necroptosis process, a variety of cytokines, cell constituents, and damage-associated molecular patterns (DAMPs) are released [71]. By stimulating the NLRP3 inflammasome and the NF-kappa B pathway, these substances induce a potent inflammatory response and immune inhibition, which further promotes tumor development and spread [72].

### 3.4. Cuproptosis

One of the most popular areas of research in the field of cancer is cuproptosis, a recently identified method of programmed cell death. Angiogenesis, proliferation, and metastasis in cancer are all linked to copper ion buildup [73]. Higher concentrations of copper are often seen in the blood or tumor tissues of patients with different cancers. Due to its dual effects of suppressing and stimulating tumor growth, copper has been known for its multifaceted role in cancer [74]. For example, a high amount of copper can lead to cell death, especially in the mitochondria, by a process called “cuproptosis”. The term “cuproplasia” refers to the need for copper in the proliferation of cancer cells [74]. Cuproptosis-related long noncoding RNAs (CRLs) and their prognostic importance in OSCC, however, are still poorly known [75]. For OSCC, focusing on it has emerged as a promising therapy approach [76].

Nine cuproptosis-related lncRNAs (CRLs) were found to be connected to the prognosis and subtypes of OSCC in the study conducted by Gong et al. [75]. The identified circular RNA ligands (CRLs) include *FAM27E3*, *MYOSLID*, *AC107027.3*, and *AC019080* as risk genes, while *AC008011.2*, *AC005785.1*, *AC020558.2*, *AC025265.1*, and *LINC02367* are classified as protective genes. These CRLs are associated with varying survival outcomes, TME characteristics, and mutation profiles, demonstrating significant predictive value in OSCC. Within the TME, two distinct clusters were identified: Cluster 1, associated with poor prognosis, exhibited higher gene alterations, greater tumor heterogeneity, and increased infiltration of M0 macrophages and resting memory CD4 T cells. In contrast, Cluster 2 was characterized by enhanced immunological activity, improved survival rates, and greater infiltration of CD8 T cells. In preclinical models of OSCC, knockdown gene expression of *FAM27E3*, *MYOSLID*, and *LINC02367* revealed reduced cell proliferation, colony formation, migration, and invasion, indicating their impact on OSCC cell characteristics. The CRLs-based signature was found to be very stable across subgroups of OSCC patients, indicating its potential for use in clinical settings regarding survival prediction. This work shows a possible signature for precision therapy and clarifies the involvement of CRLs in the TME of OSCC.

Another Study by Li et al. [77] identified seven cuproptosis-related lncRNAs, with C6orf99, AC010894.2, AC099850.4, and RPL23AP7 as risk genes, and AL513190.1, AC098484.2, and AC09587.2 as protective genes. In the low-risk group, differentially expressed genes included *KMT2E*, *CACNA1C*, *ATRX*, *PIK3CA*, *TPTE*, *LRRTM1*, *ASXL3*, and *USP34*. In contrast, the high-risk group exhibited interactions among mutations in *CDKN2A*, *TP53*, *AJUBA*, and *NEB*. Immune checkpoint genes such as *PD-1* and *CTLA4* were expressed at lower levels in the low-risk group, suggesting a potentially enhanced response to immunotherapy. Additionally, compared to the low-risk group, there were fewer Tgd cells and neutrophils, while levels of aDC, pDC, B cells, CD8 T cells, cytotoxic cells, TIL, Treg, Th, Th2 cells, Tcm, Tfh, and mast cells increased [77].

A recent study identified eight cuproptosis-related lncRNAs (CRLs): *THAP9-AS1*, *WDFY3-AS2*, *AL132800.1*, *GCC2-AS1*, *LINC00847*, *STARD4-AS1*, *CDKN2A-DT*, and *AC005746.1* [76]. In the high-risk group, risk genes such as *THAP9-AS1*, *WDFY3-AS2*, *LINC00847*, *AL132800.1*, and *GCC2-AS1* were found to be up-regulated, while protective genes *STARD4-AS1* and *AC005746.1* were down-regulated. Moreover, there was a greater presence of tumor-associated macrophage M2 (TAM M2), myeloid-derived suppressor cells (MDSC), and cancer-associated fibroblasts (CAFs) in the high-risk cohort. Survival analysis that combined tumor mutational burden (TMB) and risk scores indicated significant differences in median survival times; the low TMB and low-risk group demonstrated the best prognosis, whereas the high TMB and high-risk group experienced the worst outcomes. Additionally, the high-risk group showed increased sensitivity to several treatments, including AKT inhibitor VIII, AZ628, BAY61-3606, embelin, epothilone B, gemcitabine, GSK-650394, imatinib, mitomycin C, MS-275, PAC-1, pyrimethamine, roscovitine, salubrinal, sorafenib, and thapsigargin, reflected by lower half-maximal inhibitory concentration (IC50) values. In contrast, erlotinib, lapatinib, WZ-1-84, and Z-LLNe-CHO displayed higher IC50 values.

Together, the three investigations demonstrate the identification of cuproptosis-related long non-coding RNAs and their prognostic importance in OSCC [75,76,77]. Based on the distinct mutation profiles, survival patterns, and other characteristics of each study, it was possible to identify CRLs as prognostic biomarkers for patient outcomes. When taken as a whole, these investigations improve our knowledge of the functions of CRLs in OSCC and highlight the significance of these proteins for targeted treatments and survival prediction.

### 3.5. Pyroptosis

Pyroptosis is a type of prearranged cell death that happens when caspase cleaves members of the gasdermin family. A growing body of evidence suggests a link between the formation of tumors and pyroptosis [78]. Pyroptosis may contribute to the growth of cancer in two ways. In contrast, numerous molecular pathways and pro-inflammatory agents produced during pyroptosis are closely associated with the event of tumorigenesis and chemotherapy resistance. Nevertheless, pyroptosis, a type of programmed cell death, can prevent cancer cells from proliferating [79]. 

Recent bioinformatics analyses have revealed an intriguing connection between pyroptosis-related gene signatures and immune cell infiltration in OSCC patients. These studies suggest that such signatures may have predictive value in identifying patients who are more likely to respond favorably to immunotherapy approaches [80,81,82]. Research in this area is currently ongoing, and there is evidence that suggests pyroptosis may be implicated in the development and progression of OSCC, even if the reasons and implications of its involvement in this condition are yet unknown [83].

In a study by Xin et al. [84], eight pyroptosis-related lncRNA signatures were identified for prognostic evaluation in OSCC: *ZFAS1*, *JPX*, *TNFRSF10A-AS1*, *LINC00847*, *AC099850.3*, *AC024075.2*, *AC136475.2*, and *IER3-AS1*. Patients were classified into high- and low-risk groups based on these signatures. The high-risk group exhibited elevated expression levels of *JPX*, *ZFAS1*, TNFRSF10A-AS1, LINC00847, AC099850.3, and *IER3-AS1*, which correlated with a poorer prognosis. In contrast, *AC024075.2* and *AC136475.2* were expressed at lower levels in the high-risk group, further indicating worse outcomes. The expression levels of all eight lncRNAs were significantly associated with clinicopathological grades, including tumor size, lymph node status, and clinical stages. Additionally, the high-risk group showed positive correlations with immune cells such as NK cells, Th1 cells, Th2 cells, B cells, macrophages, and T cells. This prognostic signature based on pyroptosis-related lncRNAs not only enhances study that relates to targeted therapy for OSCC patients but also has the potential to serve as an independent marker for prognosis in OSCC patients.

In a different study, three pyroptosis-related signatures (*IL12RB2*, *CD5*, and *CTLA4*) were found to be protective genes in OSCC prognosis [80]. There were found to be high and low-risk groups, with the low-risk group being linked to high expression of these three genes. While naive B cell and memory B cell, CD8, activated memory CD4 T cells, Tfh, Treg, M1 and M2 macrophages, and resting mast cells were more expressed in the low-risk group, naive CD4 T cell, monocytes, M0 macrophage, Tfh, active mast cells, and eosinophils were enhanced in the high-risk group. Higher TIDE scores in the high-risk group suggested an insufficient immune response to immunotherapies. Ultimately, the study showed that this model could reliably predict results and that these three genes were independent prognostic markers linked to OS. The study found a link between pyroptosis and OSCC, which opens new treatment options for the treatment and prevention of OSCC.

### 3.6. Disulfidptosis

Disulfide-induced cytoskeleton protein collapse and intracellular disulfide accumulation are characteristic of a unique form of disulfide-induced cell death called disulfidptosis [85]. Preclinical studies suggest that GLUT inhibitor-based metabolic therapy can induce disulfidptosis and halt the growth of cancer [86]. In conditions of glucose starvation and lacking repair mechanisms, there can be an excessive buildup of disulfides within solute carrier family 7 member 11 (SLC7A11). Cells that express high levels of SLC7A11 undergo disulfide stress, which triggers disulfidptosis, a unique form of cell death defined by a specific underlying mechanism [87]. Research suggests that using GLUT inhibitors could effectively induce disulfidptosis in tumor cells with high SLC7A11 expression, presenting a novel strategy for cancer treatment. Long non-coding RNAs associated with disulfidptosis (DRLs) have been found as possible prognostic markers and therapeutic targets in the context of OSCC, especially in HPV-negative cases. These DRLs can help predict patient prognosis, immune cell function, and immunotherapy response [88].

Yang et al. [88] successfully identified two disulfideptosis-related lncRNA signatures, *AC104794.3* and *AL109936.2*, as prognostic markers and protective genes in HPV-negative OSCC. These lncRNAs showed negative correlations with *SLC7A11*, *GYS1*, and *SLC3A2* and positive correlations with *RPN1*, *OXSM*, and *NDUFA11*. Based on these signatures, patients were categorized into high- and low-risk groups, with the low-risk group demonstrating better OS than the high-risk group. The high-risk group showed a higher prevalence of T cells, resting memory CD4 cells, resting NK cells, M0 macrophages, aDCs, and mast cells, while the low-risk group had more Tregs, CD8 T cells, Tfh cells, plasma cells, and naïve B cells. Additionally, factors such as antigen-presenting cells (APCs) co-stimulation, CD8 T cells, B cells, checkpoint signaling, iDCs, NK cells, T cell co-stimulation, Tfh cells, Th2 cells, and TILs were more abundant in the low-risk group. The high-risk group exhibited a higher TIDE score, indicating a poorer response to immunotherapy and a greater likelihood of immune evasion, with an increased TMB linked to worse outcomes, particularly in high-risk patients. In terms of drug sensitivity, the high-risk group showed reduced sensitivity to AZD3759, erlotinib, gefitinib, trametinib, PD0325901, and ulixertinib, but heightened sensitivity to sorafenib, crizotinib, MK-1775, ZM447439, talazoparib, AZD6738, VE821, and GDC0810. Bioinformatics analyses revealed that lncRNA *AC104794.3* is downregulated in OSCC in vitro preclinical models, and functional assays showed that its overexpression could obstruct OSCC cell migratory, proliferative and viable activity, suggesting its role as a tumor suppressive gene in OSCC tumorigenesis. Thus, lncRNA *AC104794.3* may serve as a potential biomarker, providing perceptions for gene therapy in OSCC in the future.

**Table 1 biomedicines-13-00134-t001:** Overview of prognostic gene signatures that are associated with programmed cell death pathways.

Authors	Findings	FDA Approved Drugs (Launch/Phase)
Li et al. [54]	Identified ferroptosis-related genes (FRGs) on the OS of OSCC patients. A total of 10 OSCC-prognostic ferroptosis-related genes were identified LASSO regression analysis (ATG5, BID, ACO1, GOT1, AKR1C3, GLS2, ALOX15, SCO2, MAP1LC3A, MAP3K5). Among these FRGs, seven genes (ATG5, BID, ACO1, GOT1, AKR1C3, GLS2, ALOX15) were regarded as risk genes, and the other three genes (SCO2, MAP1LC3A, MAP3K5) were considered protective genes. High-risk group was associated with low CD8, Natural killer (NK) cells, tumor infiltrating lymphocytes (TILs), T helper (Th) cells, Th2 cell, Treg cell. Low-risk group was associated with high activated dendritic cells (aDCs), B cells, immature DC (iDC) cells, mast cells, plasmacytoid DC (pDC), type II IFN response.	GOT1 (L-cysteine); AKR1C3 (bimatoprost, C11-Acetate, diclofenac, rutin); GLS2 (L-glutamic acid);
Hou et al. [35]	Identified gene signature based on autophagy-related genes (ARGs) that related to prognosis in OSCC. A total of 13 prognostic ARGs were identified based on LASSO regression analysis (USP10, ATF6, MAPK9, BID, FOS, MAP1LC3A, SPHK1, GRID1, IKBKB, RAB24, CFLAR, WDR45, RAF1). Among those, four genes (USP10, ATF6, MAPK9, BID) were considered as risk genes, while the other genes (FOS, MAP1LC3A, SPHK1, GRID1, IKBKB, RAB24, CFLAR, WDR45, RAF1) were considered as protective genes. Did not report the correlation with immune cell in this study	ATF6 (ephedrine-(racemic); MAPK9 (PGL5001, RGB-286638);
Huang et al. [67]	Identified six genes namely HPRT1, PGAM5, BID, SMN1, FADD, and KIAA1191 were related to necroptosis In differential expression analysis HPRT1, PGAM5, BID, SMN1 and FADD exhibited higher expression in OSCC tissues, while KIAA1191 exhibited no significant difference in expression in OSCC and normal tissues. HPRT1 and BID were identified as risk genes. Expression of HPRT1 was correlate positively with Th2 cells, Th cells and Tgd cells and negatively correlated with aDC, B cells, CD8 T cells, cytotoxic cells, DC, eosinophils, iDC, macrophages, mast cells, neutrophils, NK CD56 bright cells, NK CD56 dim cells, NK cells, pDC, T cells, Tcm, Tem, Tfh, Th1, Th17, and Treg. Expression of BID was correlate positively with NK CD56 dim cells, NK cells, Th2, macrophages and negatively correlated with aDC, B cells, CD8 T cells, cytotoxic cells, DC, eosinophils, iDC, mast cells, neutrophils, NK CD56 bright cells, pDC, T cells, Th cells, Tcm, Tem, Tfh, Tgd, Th1, Th17, and Treg. Knockdown of HPRT1 inhibited cell proliferation and migration ability as compared to control in OSCC in vitro preclinical models.	HPRT1 (azathioprine, C11-Acetate, mercaptopurine, polyinosine); SMN1 (LMI070)
Gong et al. [75]	Identified nine cuproptosis-related IncRNAs (CRLs) associated with OSCC prognosis and subtypes. The nine CRLs identified in the study are: FAM27E3, MYOSLID, AC107027.3, AC019080 as risk genes whereas AC008011.2, AC005785.1,.5, AC020558.2, AC025265.1 and LINC02367 as protective genes. These CRLs were found to have prognostic significance in OSCC and were associated with distinct survival patterns, TME and mutation profiles. Two distinct clusters were identified in exploration of the TME, Cluster 1 which associated with poor prognosis in OSCC has higher M0 macrophages and resting memory CD4 T cells infiltration, higher gene mutations, increased tumor heterogeneity and poor prognosis, while Cluster 2 shows better survival, higher immune activity, and CD8 T cell infiltration. Knockdown of FAM27E3, MYOSLID and LINC02367 inhibited cell proliferation and colony formation, migration and invasion ability as compared to control in in vitro of OSCC preclinical models.	N/A
Xin et al. [84]	Identified eight pyroptosis-related lncRNA signatures (AC024075.2, AC136475.2, JPX, ZFAS1, TNFRSF10A-AS1, LINC00847, AC099850.3 and IER3-AS1) for prognoses in OSCC. Based on eight prognostic signatures in OSCC, individuals were classified into high- and low-risk groups. In the high-risk group, the expression levels of ZJPX, ZFAS1, TNFRDF10A-AS1, LINC00847, AC099850.3, and IER3-AS1 were higher compared to those in the low-risk group (risk gene), correlating with poor prognosis. Conversely, AC024075.2 and AC136475.2 exhibited lower expression levels in the high-risk group than in the low-risk group (protective gene), also indicating poorer prognosis. The expression level of all eight lncRNA were significantly associated with clinicopathological grade including, T stage, N stage and clinical stage in OSCC patients. High-risk group exhibited a positive correlation with macrophages, T cells, B cells, Th1, Th2, and NK cells populations.	
Li et al. [77]	Identified seven cuproptosis-related lncRNAs with C6orf99, AC010894.2, AC099850.4 and RPL23AP7 as risk gene and AL513190.1, AC098484.2, and AC09587.2 as protective gene. Differentially genes included USP34, ASXL3, LRRTM1, TPTE, PIK3CA, ATRX, CACNA1C, KMT2E (low-risk) and TP53, AJUBA, CDKN2A, NEB (high-risk), showing interaction effects among these mutations. Low-risk showed lower expression of immune-checkpoint gene like CTLA4 and PD-1 suggested might have better immunotherapy response potential. Low-risk group was associated with high number of aDC, pDC, B cells, CD8 T cells, cytotoxic cells, TIL, Treg, Th, Th2 cells, Tcm, Tfh and mast cells and lower number of Tgd cells and neutrophils.	N/A
Fan et al. [55]	Identified nine prognostic-related differentially expressed ferroptosis-related genes (PR-DE-FRGs): CISD2, DDIT4, CA9, ALOX15, ATG5, BECN1, BNIP3, PRDX5, and MAP1LC3A. KM analysis showed better OS with low expression of CISD2, DDIT4, CA9, ALOX15, ATG5, BECN1, BNIP3, and PRDX5 (risk gene), except for MAP1LC3A (protective gene). Higher CISD2, MAP1LC3A, and PRDX6 expression correlated with higher grade, while CISD2, ATG5, and DDIT4 were associated with stage III–IV. High-risk scores correlated with lower immune and stromal scores, and decreased levels of various immune cells such as B cells, CD8 T cells, DCs, iDCs, macrophages, mast cells, neutrophils, pDCs, Th, Tfh, Th2, TIL and Treg. All nine PR-DE-FRGs exhibited sensitivity to specific chemotherapy drugs: ALOX15 correlated positively with doxorubicin and etoposide, BECN1 negatively with docetaxel, BNIP3 positively with cisplatin, and MAP1LC3A negatively with methotrexate.	CA9 (coumarin, curcumin, ethoxzolamide, hydrochlorothiazide, hydroflumethiazide, mafenide, saccharin, zonisamide); PRDX5 (auranofin, benzoic-acid, diminazene-aceturate)
Liang et al. [76]	Identified eight cuproptosis-related LncRNAs (CRLs): THAP9-AS1, STARD4-AS1, WDFY3-AS2, LINC00847, CDKN2A-DT, AL132800.1, GCC2-AS1, AC005746.1. Up-regulation observed in THAP9-AS1, WDFY3-AS2, LINC00847, CDKN2A-DT, AL132800.1, and GCC2-AS1 (risk genes) in the high-risk group, while down-regulation noted in STARD4-AS1, CDKN2A-DT, and AC005746.1 (protective genes). CAF, MDSC, and tumor associated macrophage (TAM) M2 were higher in the high-risk group. Joint survival analysis of TMB and risk scores showed significant differences in median survival times; low TMB combined with low-risk showed the best prognosis compared to high TMB combined with high-risk. High-risk group exhibited higher sensitivity (lower IC50) to AKT inhibitor VIII, AZ628, BAY61-3606, embelin, epothilone B, gemcitabine, GSK-650394, imatinib, mitomycin C, MS-275, PAC-1, pyrimethamine, roscovitine, salubrinal, sorafenib, thapsigargin and higher IC50 for erlotinib, lapatinib, WZ-1-84 and Z-LLNe-CHO.	N/A
Qi et al. [80]	Identified three pyroptosis-related signatures: CTLA4, CD5, and IL12RB2 (protective gene) in prognosis of OSCC. High- and low-risk groups were identified, and it was discovered that the high expression of these three genes was associated with low-risk group. In the high-risk group, naive CD4 T cell, monocytes, M0 macrophages, activated mast cell, and eosinophils were increased, while in the low-risk group, naïve B cell, memory B cell, CD8 T cells, activated memory CD4 T cells, Tfh, Tregs, M1 macrophages, M2 macrophages, and resting mast cell exhibited higher number compared to high-risk group. The higher TIDE (Tumor Immune Dysfunction and Exclusion) scores in high-risk group, suggested the unsatisfactory response to immunotherapies of the high-risk group.	CD5 (chloramphenicol)
Yang et al. [88]	Identified two disulfidptosis-related lncRNA signatures, AC104794.3 and AL109936.2 (protective genes) as prognostic markers in HPV-negative OSCC. These lncRNAs showed positive associations with RPN1, OXSM, and NDUFA11, and negative correlations with SLC7A11, GYS1, and SLC3A2. Patients categorized into high and low-risk groups based on these signatures; low-risk group demonstrated better OS compared to high-risk. High-risk group exhibited higher levels of resting memory CD4 T cells, resting NK cells, M0, aDCs, and mast cells. Low-risk group showed higher levels of naïve B cells, plasma cells, CD8 T cells, Tfh cells, and Tregs. Low-risk group had heightened levels of APC co-stimulation, CD8 T cells, B cells, checkpoint signaling, iDCs, NK cells, T cell co-stimulation, Tfh, Th2 cells, and TILs. High-risk group showed higher TIDE score, suggesting poorer immunotherapy response and increased potential for immune escape. High TMB associated with poorer prognosis especially with high-risk status. High-risk group less responsive to AZD3759, erlotinib, gefitinib, trametinib, PD0325901, ulixertinib, and foretinib; more responsive to sorafenib, crizotinib, MK-1775, ZM447439, talazoparib, AZD6738, VE821, and GDC0810 compared to low-risk group Ectopic expression of AC104794.3 in in vitro reduced the cell proliferation and migration ability as compared to control.	N/A

N/A is not available.

## 4. Epigenetic and Gene Regulation

Epigenetics and gene regulation are essential mechanisms that control gene expression without modifying the actual DNA sequence. These processes affect cellular differentiation, development, and reactions to environmental changes, including the advancement and treatment of diseases such as cancer. Epigenetic regulation encompasses modifications like DNA methylation, histone changes, and RNA-associated processes that can either activate or suppress gene expression. Gene regulation covers a wider array of processes, including both transcriptional and post-transcriptional control facilitated by messenger RNAs (mRNAs), long non-coding RNAs (lncRNAs), RNA modifications (like N7-methylguanosine), and telomere maintenance systems such as the shelterin complex. In relation to cancer, disturbances in these regulatory pathways aid in the initiation, advancement, and spread of tumors. For instance, mRNAs and their expression patterns are vital biomarkers for prognosis and potential therapeutic targets in OSCC. Likewise, stemness indices based on mRNA expression assist in categorizing tumor subtypes and forecasting immune cell infiltration. Modifications to RNA, such as N7-methylguanosine, are crucial for stabilizing and regulating RNA transcripts, thereby affecting gene expression and responses to therapy. The shelterin complex, which is essential for telomere maintenance, plays a key role in chromosomal stability and the longevity of cancer cells. Another significant class of regulatory elements, long non-coding RNAs, affects chromatin remodeling, mRNA stability, and immune modulation, presenting new avenues for treatment interventions. The integration of these epigenetic and gene regulatory factors offers a thorough framework for comprehending cancer biology and crafting personalized treatment approaches. This summary lays the groundwork for examining the specific functions of mRNAs, mRNA expression-based stemness indices, telomere maintenance, RNA modifications, and lncRNAs in OSCC.

### 4.1. mRNA

Messenger RNAs (mRNAs) have garnered significant interest in cancer research for their potential as predictive indicators and their important roles in various functional and cellular mechanisms [89]. Additionally, studies have indicated that an incorporated model comprised of numerous genes serves as a more promising biomarker compared to relying on a single clinical biomarker [90].

Cao et al. [37] successfully predicted that a three-mRNA signature consists of *CLEC3B*, *C6*, and *CLCN1* genes as a prognostic biomarker pattern for OSCC according to the TCGA OSCC transcriptomic database. Using univariate Cox regression analysis, 120 DEmRNAs were found to have prognostic significance. After LASSO filtered out with the balance of 50 genes, stepwise multivariate Cox regression analysis was used to create the final selection of the three-mRNAs. This three-mRNA signature demonstrated enhanced 3- and 5-year OS performance along with a moderate degree of predictive capacity. When compared to other sociodemographic and clinic-pathological variables, including gender, age, race, survival status, pathological stage, tumor grade, and tumor, node, metastasis (TNM) stage, it is shown to be the most significant and independent determinant for the risk of OSCC survival. According to this study, downregulation of the tumor suppressor genes *CLEB3B*, *C6*, and *CLCN1* is associated with a poor prognosis. Inhibitors of cell growth are favorably correlated with *CLEB3B*, which lowers cell proliferation in OSCC [91]. Comparably, it was found that downregulated *CLCN1* also indicated a bad prognosis, which may have happened because of chloride channels’ role in controlling the cell cycle [92]. 

Another study by Feng et al. [93] identified mRNA based-stemness index known as mRNAsi-related gene signature in OSCC. The TCGA cohort’s mRNAsi was quantified, and both low- and high-mRNAsi groups’ prognosis and TME were assessed. The ESTIMATE method was used to evaluate the immunological and stromal scores. The high-mRNAsi group showed higher immunological and stromal scores than the low-mRNAsi group. Additionally, there were more CD8 T cells, activated memory CD4 T cells, and resting NK cells infiltrating the high-mRNAsi group. On the other hand, the group with low mRNAsi showed increased infiltration of CD4 T cells that are resting memory. A combination of univariate and LASSO Cox regression was used to calculate OSCC survival-related DEGs. Lastly, a risk score model was created using multivariate Cox analysis. A total of eleven mRNAsi-associated signature genes were identified, with a high-risk score being significantly associated with a reduced likelihood of survival. While there were no noticeable differences in the levels of *TPSAB1*, *CCL22*, and *TSPAN11* between OSCC and normal tissues, the study found upregulation of *CCDC92*, *KPNA2*, *NPM3*, *TWIST2*, *H2AFZ*, and *GAS1*, along with downregulation of *CLEC3B* [93]. Notably, elevated expression of *H2AFZ* has been observed in several cancer types, including breast cancer and hepatocellular carcinoma [94,95]. This investigation further confirmed that OSCC displays higher levels of *H2AFZ* expression compared to normal tissues. In patients with OSCC, the remaining mRNAsi-associated genes were investigated further. According to the findings, there was a significant survival advantage for individuals with higher levels of *CLEC3B*, *CCL22*, *TPSAB1*, *TWIST2*, *IGLV2*, *GASI*, *CCDC92*, and *TSPAN1*. On the other hand, overexpression of *H2AFZ*, *KPNA2*, and *NPM3* was linked to a worse prognosis [93]. These results imply that the stemness index-related signature may serve as a valuable prognostic indicator and help with OSCC risk classification.

### 4.2. mRNA Expression-Based Stemness Index

Another subset of RNA known as lnRNA was reported to be involved in various biological processes through interaction with protein, RNA, and chromatin [96]. N6-methyl adenosine, known as m6A, is a prevalent epigenetic methylation modification that occurs on mammalian RNA, including lncRNA [97]. The process of m6A modification on lncRNA involves different protein factors that are categorized into writers (methyltransferases), readers (signal transducers), and erasers (demethylases) [98]. Correlation analysis can be used to identify the lncRNAs associated with m6A genes. The role of m6A-related lncRNAs in tumor metabolism, as well as a prognosticator in a few cancers, including renal carcinoma, colorectal cancer, lung cancer, and glioblastoma, were reported in published literature [99,100,101]. Both mRNA and lncRNA are documented as being involved in the initiation and prognosis of OSCC [102,103]. Yang et al. [103] identified 271 m6A-related lncRNA, of which 16 of them (*AC079684.2*, *AC092115.4*, *LINC01644*, *LINC01410*, *AL355574.1*, *AC091271.1*, *LINC00630*, *ALMS1-IT1*, *LINC00992*, *AC099850.4*, *AC005288.1*, *AC107027.3*, *JPX*, *LINC01775* and *PTOV1-AS1*) were strongly correlated with OSCC survival outcomes were identified by univariate Cox regression analysis. All of them were regarded as a risk gene except *LINC01775* as a protective gene. Immune signatures, TME, and metastasis gene expression between two clusters were identified via the CIBERSORT and ESTIMATE algorithm. The high-risk group exhibited low immune scores. There was a negative correlation between naive B cells and resting memory CD4 T cells in tumor infiltration and risk scores. Using LASSO regression, 11 m6A-related lncRNAs prognostic signature (m6A-RLPS) were chosen to create a risk prediction signature, m6A-RLPS (*LINC01644*, *LINC01410*, *AL355574.1*, *AC091271.1*, *LINC00630*, *LINC00992*, *AC099850.4*, *AC005288.1*, *AC107027.3*, *JPX*, *LINC01775*). Results showed that the m6A-RLPS have a strong competence to predict the OS of OSCC patients by risk score calculation. Therefore, suggests that the risk signature has the potential to not only aid in the prognostic prediction and treatment of OSCC but also facilitate further research in OSCC.

### 4.3. Telomerere (Shelterin Complex)

Telomere maintenance is necessary to support the unabated growth of cancer, which is why telomeres are seen as universal anti-cancer targets. In cancer, one frequent mutation observed involves telomerase, the enzyme responsible for maintaining telomeres [104]. To preserve chromosomal ends, a crucial protein complex known as shelterin controls telomerase connections, stops telomere erosion and avoids improper DNA repair. Due to their impact on tumor growth and metastasis, changes in shelterin components can affect cellular longevity, hasten the evolution of cancer, and serve as important targets for cancer therapy [105]. To completely comprehend their role and develop more effective treatment approaches, more research is necessary. Shelterin genes may possess a significant prognostic influence on the course of OSCC [106].

A study conducted by Zhang et al. [106] found a predictive pattern for shelterin complex genes (SGs) in an extensive analysis of OSCC, indicating that SGs have a major influence on TME and disease outcomes. They identified *TRF1*, *TRF2*, *RAP1*, and *POT1* were linked to increased risk and a worse prognosis in patients with OSCC. Simultaneously, *TIN2* became a good prognostic indicator. Compared to normal tissues, OSCC showed elevated expression of *TRF1*, *TRF2*, *RAP1*, and *POT1*, but normal tissues showed higher levels of *TIN2*. The study used LASSO regression analysis to build a shelterin complex-related predictive model utilizing TCGA and GEO databases. This model was verified across randomized cohorts and a variety of clinical subgroups. TMB, TME traits, and clinical factors were used in this model to differentiate between high-risk and low-risk groups. Lower TMB and a markedly increased infiltration of immune cells, such as aDC, B cells, CD8 T cells, co-stimulating APCs, and other immunological components, were associated with the low-risk group. Higher immunotherapy prediction scores (IPS) and enhanced CTLA4 expression in the low-risk group were indicative of these characteristics, which were associated with improved prognosis and increased immunotherapy response. The low-risk group demonstrated lower IC50 values for several medications, including metformin, methotrexate, gefitinib, and rapamycin, according to drug sensitivity studies, indicating increased sensitivity to these treatments. On the other hand, their IC50 values for medications such as cisplatin, midostaurin, and pazopanib were higher, indicating a decreased sensitivity. The results demonstrate the potential of SGs as new treatment targets and prognostic indicators for OSCC. This research intends to improve overall treatment results for OSCC patients, improve prognosis prediction, and customize immunotherapy methods by incorporating SG-related risk ratings into clinical practice.

### 4.4. RNA Modification (N-7-Methylguanosine)

N7-methylguanosine (m7G) is an important alteration located at the 5′ cap of RNA that is involved in splicing, translation, transcription elongation, and mRNA stability [107]. RNA modifications, including m7G, are vital for regulating gene expression at both transcriptional and posttranscriptional levels [108]. Although recent studies emphasize the importance of m7G-related lncRNAs in tumor prognosis (36333719, 36468039, 36935487), research specifically focusing on m7G-related lncRNAs in OSCC is lacking [109,110,111,112]. Wang et al. [112] study explore the prognostic impact of 5 m7G-related lncRNAs in OSCC, identifying AC108488.3 as a protective gene and *AL133444.1*, *AL359091.4*, *AC007128.1* and *AL162413.1* as risk factors. The study demonstrated the differential expression of these lncRNAs and their importance in predicting OSCC prognosis and OS rates by using RNA sequencing data from TCGA. The study developed a risk score model based on these m7G-related lncRNAs, which demonstrated that high-risk groups were more responsive to doxorubicin therapy and had lower IC50 values than the low-risk group. Furthermore, immune cell infiltration, particularly of CD8 T cells, was observed to be enhanced in high-risk patients, while APC co-inhibition, cytolytic activity, and HLA response were downregulated in low-risk individuals. The combination of TMB and risk scores showed substantial relationships with OSCC survival, even though the mutation frequencies and TMB scores of the high- and low-risk groups were similar. The results imply that m7G-related lncRNAs play a critical role in therapy responsiveness and OSCC prognosis. To better understand and utilize these lncRNAs in clinical settings and investigate their functional roles in the course and therapy of OSCC, future research should fill in these gaps.

### 4.5. LncRNA

Long non-coding RNAs (lncRNAs) featuring N7-methylguanosine modifications constitute a heterogeneous class of regulatory molecules that employ a wide array of mechanisms to influence gene expression. These specialized lncRNAs demonstrate unique expression profiles within neoplastic tissues and have been implicated in the process of cellular transformation from normal to malignant states. Their involvement extends to various stages of cancer development and progression [113]. Practically every facet of gene regulation is impacted by these molecules, including chromatin remodeling, mRNA production, protein signaling, and the regulation of both innate and adaptive cellular immune responses. A study has connected several lncRNAs, such as LINC01137 and LINC01929, to the prognosis of patients with OSCC [114].

A 14-long non-coding RNA (lncRNA) profile was found by Xu et al. [114] to be a possible predictive biomarker for OSCC. This signature comprises protective genes such as *LINC00567*, *LINC00877*, *KANSL1-AS1*, *LINC01191*, *LINC00689*, and *LINC01281*, while risk genes include *WDFT3-AS2*, *NPSR1-AS1*, *ALMS1-IT1*, *HLA-F-AS1*, *LINC-PINT*, *LINC00958*, *Aflap-AS1*, and *PRKG1-AS1*. The prognosis for the low-risk group was substantially better than that of the high-risk group. More NK cells, neutrophils, pDCs, aDCs, eosinophils, macrophages, mast cells, monocytes, Th1, Th17 cells, activated B cells, activated CD8 T cells, CD8 Tem, Tgd, immature B cells, Treg, Tfh, Th1, and Th17 cells were associated with the low-risk grouping. Nonetheless, the high-risk group had fewer of these immune cells. Negative relationships between the SRG signature and activated B cells, monocytes, macrophages, active CD8 T cells, Tfh, and MDSCs were discovered. By using LASSO analysis, multivariate Cox regression, and univariate Cox regression on RNA sequencing data from TCGA and GEO, the study created and validated the lncRNA-based prognostic signature. Receiver operating characteristic (ROC) curves, calibration curves, and Kaplan–Meier survival analysis were used to confirm the effectiveness of the signature. Moreover, ssGSEA analysis demonstrated differences in immune cell infiltration between risk groups, underscoring the potential of this 14-lncRNA signature in guiding immunotherapy approaches for OSCC.

**Table 2 biomedicines-13-00134-t002:** Overview of prognostic gene signatures that are associated with epigenetic and gene regulation pathways.

Authors	Findings	FDA Approved Drugs (Launch/Phase)
Cao et al. [37]	Identified a three-mRNA signature that is associated with the OS of OSCC patients. A three-mRNAs were identified as a prognostic biomarker pattern for OSCC (CLEC3B, C6 and CLCN1). These three-mRNA signature were regarded as risk genes and may serve as tumor suppressor in OSCC, and the downregulation of these mRNAs in OSCC indicated poor prognosis. Did not report the correlation with immune cell in this study.	CLCN1 (fenobibric-acid, nifllumic acid)
Feng et al. [93]	Identified an mRNA expression-based stemness index- (mRNAsi-) associated signature in OSCC. A total of 11 genes were identified, and high-risk score was related to poor survival outcomes. H2AFZ, KPNA2 and NPM3 are regarded as risk genes, CCDC92, GAS1, IGLV2, TWIST2, CLEC3B, CCL22, TPSAB1 and TSPN11 are regarded as protective genes. High-risk group was associated with increased levels of CD8 T cells, activated memory CD4 T cells, resting NK cells Low-risk group was associated with higher level of resting memory CD4 T cells, Treg and T cells gamma delta (Tgd) populations. Involved functional analyses to investigate the oncogenic role of H2AFZ gene in in vitro preclinical models of OSCC.	N/A
Yang et al. [103]	Identified m6A-related lncRNA related to OSCC prognosis. A total of 16 m6A-related lncRNAs were identified to be highly correlated to OSCC survival outcomes. OSCC samples were divided into Cluster 1 and Cluster 2 based on the differential expression of 16 m6A-related lncRNAs where Cluster 2 was associated with poor prognosis. Cluster 1 was significantly associated with high number of naïve B cells, resting NK cells and DCs whereas Cluster 2 was associated with high number of resting memory CD4 T cells. A total of 11 m6A-related lncRNAs were selected from prognostic lncRNAs to establish m6A-RLPS (LINC01644, LINC01410, AL355574.1, AC091271.1, LINC00630, LINC00992, AC099850.4, AC005288.1, AC107027.3, JPX, LINC01775). All are regarded as risk genes except LINC01775 as protective gene.	N/A
Zhang et al. [106]	Identified shelterin complex-related signature related to prognosis and TME in OSCC. Six shelterin complex genes (SGs) were significantly elevated in OSCC group (TRF1, TRF2, RAP1, TPP1, POT1, TIN2). Four were identified as OSCC risk factor (TRF1, TRF2, ACD (RAP1) and POT1) except TIN2 identified as favorable factor. Low-risk was associated with lower TMB and exhibited more abundant immune cell infiltration aDCs, co-stimulation antigen presenting cell (APC), B cells, chemokine receptor, CD8 T cell, checkpoint, cytolytic activity, HLA, iDC, mast cells, neutrophil, NK cells, pDC, T cell co-inhibition/stimulation, Th cell, effector T cell, Th1, Th2 and TIL. IPS score was higher in low-risk group that indicate these patients received immunotherapy more readily than high-risk patients. Low-risk group has higher expression of CTLA4. Low-risk group has lower half-maximal inhibitory concentration (IC50) for metformin, methotrexate, gefitinib, rapamycin, nilotinib and high IC50 for pazopanib, midostaurin, bexarotene, imatinib, docetaxel, dasatinib, cisplatin, sorafenib, bicalutamide, gemcitabine, and embelin.	N/A
Wang et al. [112]	Identified five m7G-related lncRNAs associated with prognosis in OSCC. These lncRNAs are: AL133444.1, AC007128.1, AL359091.4, AL162413.1 as risk gene and AC108488.3 as protective gene. The expression of the five m7G-related lncRNAs was significant in predicting prognosis and OS rates in OSCC patients. High-risk groups were sensitive to Doxorubicin treatment with lower IC50 compared to low-risk group. High-risk group had high level of immune cells infiltration, especially CD8 T cells. Low-risk groups had downregulated APC co-inhibition, cytolytic activity, and HLA response pathways. The mutation frequency and TMB score did not significantly differ between high-risk and low-risk patient groups. However, when combining TMB with the risk score analysis, a significant correlation with OSCC survival was observed.	N/A

## 5. Metabolism and Energy

Metabolism and energy control are vital for maintaining cellular balance and facilitating crucial functions such as growth, repair, and adaptation to environmental shifts. In contrast, cancer cells manipulate these metabolic pathways through a phenomenon known as metabolic reprogramming to enhance their rapid growth and survival. This reprogramming entails a change in nutrient usage, energy generation, and metabolite conversion, resulting in modified cellular signaling and interactions with the microenvironment. Significant metabolic changes encompass boosted glycolysis, heightened lactic acid production, and modified metabolism of amino acids, lipids, and polyamines. These alterations also influence the TME, promoting immune suppression and resistance to therapies. In OSCC, metabolic reprogramming has been associated with tumor development, prognosis, and evasion of the immune response. Different metabolic subtypes include various pathways like glycolysis, lysosomal function, polyamine metabolism, and lactic acid synthesis. For example, glycolysis not only produces energy but also generates lactic acid, which transforms the TME into an environment that suppresses the immune response. Lysosomes, recognized for their role in breaking down cellular components, play a part in drug resistance and tumor invasion. Additionally, the dysregulation of polyamine metabolism, crucial for cell division, presents new therapeutic targets. Likewise, the buildup of lactic acid in OSCC influences both tumor development and the infiltration of immune cells. By combining metabolic signatures with gene expression data, researchers have created prognostic models for OSCC that help identify patients at high-risk and shape personalized treatment approaches. These metabolic pathways provide promising opportunities for targeted therapies, potentially leading to better outcomes in cancer treatment.

### 5.1. Metabolic Pathways

Cell metabolism refers to the complex network of biochemical processes that transform metabolites to support essential biological functions [115]. In normal conditions, basic metabolites and signaling pathways work together to maintain cellular balance and function, and many studies have shown that cell metabolism is significantly altered in cancerous cells [116,117]. As one of the crucial events in cancer progression, changes in the activity of this pathway could accelerate tumor growth, and this has sparked considerable interest in the study of tumor metabolism [118,119]. Thus, metabolic genes have garnered extensive attention from many cancer researchers, leading to numerous related studies such as prostate cancer, gliomas and hepatocellular carcinoma [120,121,122]. The specific mechanism between cancer and metabolic reprogramming has been extensively studied, but the specific mechanism underlying this process in OSCC remains poorly understood.

A study by Wu et al. [36] identified a transcriptional prognostic signature from five metabolic pathways in OSCC. RNA-seq data of OSCC were obtained from TCGA and GEO databases and were transformed into a metabolic pathway enrichment score matrix by ssGSEA. Finally, via an integrative bioinformatic approach, a novel prognostic signature based on a metabolic pathway known as 5-metabolic pathways (5-MPS) prognostic signature consisting of hsa00561 (glycerolipid metabolism), hsa00534 (glycosaminoglycan biosynthesis-heparan sulfate/heparin), hsa01230 (biosynthesis of amino acid), hsa00910 (nitrogen metabolism) and hsa00670 (one carbon pool by folate) was developed and validated. Validation of 5-MPs predictive values was performed in multiple independent OSCC cohorts, where patients in the high-risk subgroup had a lower OS ratio compared to the low-risk subgroup. Based on univariate and multivariate Cox regression analysis, 5-MPS was correlated with OS in OSCC patients. A total of five representative genes were selected to characterize these metabolic pathways integrated into 5-MPS, such as *CA9*, *DGKG*, *EXTL2*, *TYMS* and *PGAM1* from these dysregulated metabolic pathways. These representative genes were highly expressed in OSCC compared to their counterparts and correlated with unfavorable survival. This aligns with past studies, which have identified that *CA9*, *EXTL2*, *PGAM1* and *TYMS* were dysregulated across various human cancers and closely linked with carcinogenesis by functioning as key enzymes underlying metabolism. In addition, high expression of *CA9* and *PGAM1* were correlated with poor prognosis in OSCC.

### 5.2. Glycolysis

The energy metabolism and biosynthesis pathways of tumors are significantly influenced by tumor metabolic reprogramming [123,124]. Glycolysis is the hallmark metabolic feature during tumorigenesis [125,126]. In addition, tumor cells possess a distinct energy metabolism pathway known as glycolysis. This metabolic process alters the way tumor cells consume energy by secreting a significant amount of lactic acid through aerobic glycolysis. As a result, an accumulation of lactic acid occurs, therefore modifying the microenvironment where the tumor cells are present [127].

The relevance and function of lncRNAs in OSCC’s glycolytic activity are poorly understood. Wu and colleagues [128] evaluated the prognostic impact of various cancer hallmarks in OSCC using the ssGEA algorithm and univariate Cox analysis. Findings identified adipogenesis, cholesterol homeostasis, glycolysis, hypoxia, MYC targets, and oxidative phosphorylation as risk factors for OSCC. However, the most significant risk factor was determined to be glycolysis according to hazard ratios and *p*-values. A combination of statistical analyses, including univariate Cox regression, LASSO regression and multivariate Cox regression analyses, were performed to construct a prognostic signature related to glycolysis with a total of ten risk genes and one protective gene determined [128]. In addition, the data from ssGSEA and TIMER databases were used to evaluate the infiltration of immune cells in OSCC. As a result, a 4-lncRNA glycolysis-related signature was developed consisting of *AL035458.2*, *LINC01281*, *AC245041.2*, and *DDN-AS1*, which can differentiate between high-risk and low-risk patients. This signature was determined to be an independent risk factor for OSCC prognosis. Analysis of the correlation between immunity and the risk signature demonstrated that low-risk group patients had higher levels of immune infiltration.

Liu et al. [129] reported another prognostic model for differentiating OSCC patients with varying treatment outcomes by using glycolysis-immune-related genes. Unlike the study by Wu et al., Liu and colleagues [129] used mRNA expression profile and corresponding clinical data taken from TCGA. In addition, datasets from GEO containing gene expression profiling and detailed survival data were also collected. These data were used to conduct differential expression analysis and functional enrichment analysis. Total of six genes were selected, including *ALDOC*, *HRG*, *IGSF11*, *MIPOL1*, *PADI3*, and *VEGFA*. Thus, the prognostic risk score was established using these six genes to distinguish patients with varying prognoses. Patients in both the TCGA and GEO databases were divided into high-risk and low-risk groups, by which the high-risk group had a lower survival rate compared to the low-risk group. The present finding is in concordance with previous studies where glycolysis-related genes and immune-related genes have demonstrated their prognostic relevance in pancreatic ductal adenocarcinoma and renal papillary cell carcinoma, respectively [130,131].

### 5.3. Metabolism

The fundamental concept of cancer metabolism is that cancer cells exhibit distinct metabolic activity when compared to normal cells [132]. Many variables can impact metabolic reprogramming, which is thought to be a defining characteristic of cancer and progression. It includes variations in how various nutrients are used, the unique needs of the cell, the tissue of origin, possibly carcinogenic and epigenetic modifications, and interactions between cells and the conditions inside the TME [126,133,134]. But there is also metabolic interaction between many cell types in the TME, which cancer cells exploit to keep growing in unfavorable environments [135]. Mutations in metabolism-related genes (MRGs) dictate the modifications in metabolic pathways that occur during carcinogenesis, progression, and metastasis [136]. Nevertheless, the creation of MRG prognostic OSCC models has not received much attention and needs more investigation that may assist in determining the prognosis of OSCC.

Zhang et al. [137] developed a prediction model for OSCC using metabolism-related genes (MRGs) sourced from TCGA. They identified 11 MRGs associated with OSCC prognosis through LASSO Cox analysis: *HPRT1*, *POLD2*, *ADA*, *GOT1*, *SHMT2*, *HADHB*, *POLE3*, *MGST1*, *ADK*, *ATIC*, and *GNPDA1*. While *GNPDA1* was found to be protective, the other MRGs were linked to unfavorable outcomes. According to the model, the immune cell profiles of high-risk and low-risk groups showed significant differences, with a higher percentage of Th2 cells and lower concentrations of Th17, DC, mast cells, and iDC. This alteration in immune cell distribution may contribute to a more aggressive form of OSCC in the high-risk group. Overall, the MRG-based approach provides valuable insights for personalized forecasting and clinical decision-making in OSCC.

### 5.4. Lysosome-Related Prognostic Signature (LRGs)

Lysosomes significantly influence the development and progression of various diseases by altering their degradation capacity and releasing contents, which contributes to increased catabolic activity in tumors [138]. Lysosomes exhibit significant alterations in volume, composition, distribution, and enzyme activity in aggressive cancer cells, which promote drug resistance and cancer growth and spread. They are more vulnerable to targeted cell killing because of their weaker membranes [139,140]. New therapeutic and diagnostic targets are needed, as evidenced by the paucity of sensitive diagnostic markers for early-stage OSCC. Lysosomes influence several activities, including apoptosis, cell signaling, and metabolism, and they are essential to the formation of OSCC. It has been demonstrated that lysosomal function inhibition can improve the efficacy of therapies for HNSCC, such as C6-ceramide nanoliposome (CNL) [141]. Furthermore, Kif5b, a lysosomal transporter connected to unfavorable outcomes in OSCC, may be a useful target for therapy [142].

The identification of a novel lysosome-related prognostic signature (LRGs) for OSCC by Liu et al. [143] has important ramifications for forecasting patient outcomes and directing treatment approaches. The 12 lysosome-related genes in this signature are *NEU1*, *CD164*, *SFTPB*, *TMEM192*, *GPC1*, *SLC46A3*, *MANBA*, *BGN*, *TPP1*, *BRI3*, *TMEM175*, and *SDCBP*. The correlation of these genes with decreased OS rates in OSCC patients suggests their significance in disease progression. In the high-risk group of OSCC patients, there were higher levels of resting memory CD4 T cells and M0 and M2 macrophages, while levels of M1 macrophages, CD8 T cells, activated memory CD4 T cells, and activated NK cells were lower. These changes in immune cell composition indicate a more immunosuppressive environment in high-risk patients, potentially worsening their condition and contributing to treatment resistance. Moreover, iDC, CD8 Tcm cells, Tgd cells, NK cells, Treg cells, and memory B cells were more prevalent in high-risk individuals, according to ssGSEA. In the TME, the team also mapped the expression profiles of these 12 LRGs and found several patterns: Gene expression analysis revealed distinct cellular patterns: *SLCA46A3*, *MANBA*, *BRI3*, and *TMEM175* were primarily found in macrophages, while GPC1 was largely restricted to epithelial cells. CD164 showed broad expression across various cell types, and *TMEM192* was specifically detected in immune cells. These results imply that genes associated with lysosomes have a role in the intricate interactions that occur within the TME, affecting the immune system as well as tumor development. The identification of this 12-LRG signature provides valuable information for prognosticating OSCC and opens new avenues for targeted therapy. The roles of a few lysosome-related genes, such as *SLC46A3*, *NEU1*, *SDCBP*, and *BRI3*, are still not well characterized, especially in OSCC. Therefore, additional study is necessary to clarify their precise roles and processes in OSCC formation since this could result in the development of novel therapeutic approaches that target lysosomes.

### 5.5. Metabolism (Polyamine)

Putrescine, spermidine, and spermine are naturally occurring mammalian polyamines that are essential for both malignant and normal cell division and function [144]. Many neoplastic diseases, including cancer, have dysregulated polyamine metabolism. There is interaction between polyamine metabolism and carcinogenic pathways, including the mTOR and RAS pathways, and higher polyamine levels in a variety of cancer types [145]. Salivary metabolism has been the subject of recent research on polyamines and OSCC [146]. Ohshima et al. [147] observed significant changes in polyamine metabolism among several metabolites in saliva samples. These findings imply that polyamine-related genes and metabolites may improve our knowledge of and ability to treat OSCC in clinical settings. Targeting polyamines is a viable strategy for expanding the field of immunotherapy and getting beyond the drawbacks of systemic medicines. Therefore, it is critical to comprehend and assess the relationships and mechanisms involving immunity, polyamines, and OSCC [148].

A recent study by Tang et al. [148] discovered that six polyamine-related differentially expressed genes (PARDEGs) affect the prognosis of OSCC: The risk gene is *RIMS2*, and the protective genes are *RIMS3*, *TRAC*, *FMOD*, *CALML5*, *SPINK7*, and *CKS2*. Among these, OSCC has high expression levels of *CKS2* and *RIMS3*. After analyzing the immune cell profiles, it was found that the high-risk group had more memory B cells, M0 macrophages, activated mast cells, and resting NK cells but fewer activated memory CD4 T cells, CD8 T cells, Tgd cells, and naïve B cells, resting mast cells, and resting DC. The study explored the connections between PARDEGs and various chemotherapeutic drugs, focusing on six specific genes. The analysis identified strong positive correlations between these genes and drug sensitivity: *SPINK7* with bendamustine, *RIMS3* with nelarabine, *CKS2* with hydroxyurea, *CALML5* with idarubicin, and *FMOD* with ABT-199 (venetoclax). These findings suggest that the expression of these genes may influence the effectiveness of the corresponding drugs. The *TTN*, *FAT1*, and *TP53* genes all showed higher mutation frequency in the high-risk group. Notably, the IC50 values for doxorubicin, paclitaxel, docetaxel, and cisplatin were lower in the high-risk group than in the low-risk group. This model identifies novel therapeutic targets for improved patient management and offers significant perceptions of the mechanisms involving polyamines in OSCC. It also predicts how patients will react to chemotherapy and describes the patterns of polyamine alteration in malignancies.

### 5.6. Metabolism (Lactic Acid)

The Warburg effect explains how cancer cells produce more lactic acid by decreasing mitochondrial function and switching from aerobic glycolysis to glucose metabolism [149]. Lactic acid leads to reprogramming of respiratory metabolism, and hence, cells hydrolyze large amounts of ATP. This is because cancers and rapidly proliferating cells have glycolytic-dependent metabolism [150]. The TME’s immunosuppressive state may be preserved by this increased lactate through both H+- and lactate-mediated mechanisms. Thus, to prevent high lactic acid levels in metabolically reprogrammed tumors, targeted therapeutic is believed to be a potent therapeutic strategy [151]. 

Shen et al. [152] and associates have identified four critical genes with noteworthy prognostic implications, providing vital new insights into the function of LRGs in OSCC. *ZNF662* has been identified as a risk gene, but *CGNL1*, *VWCE*, and *ZFP42* are protective genes since they are linked to better outcomes. Subtype 2 patients, who had lower LRG scores, had significantly higher survival rates than subtype 1 patients. Subtype 1 had the greatest stromal score, indicating a more robust tumor stroma, and both subtypes had high ESTIMATE and immunological ratings, indicating active immune responses. This differentiation was also evident in differences in stromal, ESTIMATE, and immune scores. Furthermore, a positive correlation was found between elevated risk ratings and enhanced medication sensitivity. Specifically, AZ6102, venetoclax, nutlin-3a, and doramapimod showed reduced IC50 values, indicating their greater efficacy in high-risk individuals. Conversely, medications like lapatinib, ERK_6604, and PD0325901 showed decreased sensitivity in the high-risk group, as evidenced by higher IC50 values. Preclinical research on AZ6102, which inhibits tankyrase (TNKS1 and TNKS2) and stabilizes Axin to disrupt the Wnt signaling pathway, may open new treatment options. Venetoclax is a BCL-2 inhibitor that has demonstrated significant effectiveness for hematologic malignancy treatment by triggering apoptosis via the mitochondrial pathway. Although its participation in OSCC is still being explored, venetoclax may hold promise for the treatment of OSCC. These findings point to the potential for developing more customized and effective OSCC treatment regimens by utilizing LRGs and associated drug sensitivities. The results provide a crucial foundation for additional research and clinical studies that will aim to validate these findings to enhance treatment outcomes and provide OSCC patients with therapeutic alternatives. When used to treat high-risk OSCC patients, certain medications pave the way for the creation of more potent treatment plans while highlighting the necessity of additional study and clinical validation.

**Table 3 biomedicines-13-00134-t003:** Overview of prognostic gene signatures that are associated with metabolism and energy pathways.

Authors	Findings	FDA Approved Drugs (Launch/Phase)
Wu et al. [36]	Identified a transcriptional prognostic signature based on the dysregulated metabolic pathway and their prognostic significance in OSCC. A prognostic signature consisting of 5-metabolic pathways (5-MPS) were developed [ hsa00561 (Glycerolipid metabolism), hsa00910 (Nitrogen metabolism), hsa00534 (Glycosaminoglycan biosynthesis-heparan sulfate/heparin), hsa01230 (Biosynthesis of amino acids) and hsa00670 (One carbon pool by folate)] via integrative bioinformatics approach. Five representative genes in 5-MPS were selected (DGKG, CA9, EXTL2, PGAM1, TYMS) and upregulated in OSCC compared to normal counterpart. Did not report the correlation with immune cell in this study.	CA9 (coumarin, curcumin, ellagic-acid, ethoxzolamide, ferulic-acid, hydrochlorothiazide, hydroflumethiazide, indisulam, mafenide, para-toluenesulfonamide, saccharin, U-104, zonisamide); TYMS (capecitabine, carmofur, doxifluridine, enocitabine, floxuridine, ftorafur, gemcitabine, gemcitabine-elaidate, nolatrexed, pemetrexed, raltitrexed, trifluridine, trimethoprim, 5-fluorouracil, 5-FP)
Wu et al. [128]	Identified Long non-coding RNAs (lncRNA) glycolysis-related signature in OSCC. Four lncRNA glycolysis prognostic signatures were identified and built (DNN-AS1, AL035458.2, LINC01281, AC245041.2) which could distinguish high and low-risk patients. DNN-AS1, AL035458.2, and AC245041.2 are regarded as risk genes whereas LINC01281 is regarded as protective gene. All four lncRNA-encoding genes were related to abnormal activation of glycolysis in OSCC and found to be an independent prognostic risk factor for OSCC. Low-risk group is associated with higher immune infiltration including B cells, CD8 T cells, Th1 cells, Th2 cells and NK cells.	N/A
Liu et al. [129]	Identified prognostic value of glycolysis-immune-related genes on OSCC. Six glycolysis-immune-related genes were identified including ALDOC, VEGFA, HRG, PADI3, IGSF11 and MIPOL1 using LASSO Cox regression analysis. High-risk group is associated with lower OS compared to low-risk group. Low-risk group is associated with higher naïve B cells, CD8 T cells, activated memory CD4 T cells, T follicular helper cells (Tfh), Treg, monocytes, M1 macrophages, M2 macrophages and resting mast cells, whereas high-risk group is associated with M0 macrophages, DC cells and activated mast cells.	VEGFA (pidolic-acid, trometamol, vandetanib); PADI3 (L-citrulline);
Zhang et al. [137]	Identified 11 metabolic-related genes includes SHMT2, HPRT1, POLD2, HADHB, POLE3, ADK, GOT1, ATIC, MGST1, ADA as risk genes, and GNPDA1 as protective gene that associated with OSCC prognosis. High-risk group exhibited higher number of Th2 cell population and lower number of DC, iDC, mast cell and Th17 cell population as compared to low-risk group.	SHMT2 (AZD4282, MC-1, mimosin); HPRT1 (azathioprine, C11-Acetate, mercaptopurine, polyiosine); POLD2 (cladribine); ADK (ABT-702, adenosine-phosphate, ribavirin); GOT1 (L-aspartic-acid, L-cysteine, L-glutamic-acid, MC-1); MGST1 (glutathione); ADA (cladribine, dipyridamole, fludarabine, pentostatin, vidarabine)
Liu et al. [143]	Identified a novel lysosome-related prognostic signature (LRGs) associated prognosis in OSCC. A total of 12 LRGs consisted of SLC46A3, MANBA, NEU1, SDCBP, BRI3, TMEM175, CD164, GPC1, SFTPB, TPP1, BGN and TMEM192 as risk genes were linked to the lower OS of OSCC patients. High-risk OSCC patients showed elevated M0 and M2 macrophages, and resting memory CD4 T cells, with reduced M1 macrophages, CD8 T cells, activated memory CD4 T cells, and activated NK cells. Single sample gene set enrichment analysis (ssGSEA) analysis indicated higher proportions of immune cells such as iDC, CD8 Tcm cells, Tgd, NK cells, Treg, and memory B cells in the high-risk group. The expression profiles of 12 LRGs within TME revealed specific patterns: SLCA46A3, MANBA, BRI3, and TMEM175 were expressed in macrophages; CD164 ubiquitously; GPC1 in epithelial cells excluding cells; and TMEM192 in immune cells.	NEU1 (oseltamivir-phosphate)
Tang et al. [148]	Identified six polyamine-related differentially expressed genes (PARDEGs) in OSCC prognosis: CKS2 as risk gene whereas RIMS3, TRAC, FMOD, CALML5, and SPINK7 as protective genes. CKS2 and RIMS3 are highly expressed in OSCC, whereas SPINK7 shows high expression in normal tissues. CKS2 is significantly upregulated in OSCC compared to adjacent normal tissue. High-risk group was associate with low number of naïve B cells, resting mast cells, resting dendritic cells, activated memory CD4 T cells, CD8 T cells, Tfh, Tgd, and Treg cells and higher number of memory B cells, M0 macrophages, activated mast cells, and resting NK cells. The analysis linked six PARDEGs to specific chemotherapy drugs: SPINK7 with bendamustine, RIMS3 with nelarabine, CKS2 with hydroxyurea, CALML5 with idarubicin, and FMOD with ABT-199, showing high correlation. The high-risk group exhibited a higher mutation rate in TP53, TTN, and FAT1 compared to the low-risk group. IC50 for cisplatin, docetaxel, doxorubicin, and paclitaxel were lower in high-risk groups compared to low-risk groups.	TRAC (3-indolebutyric-acid)
Shen et al. [152]	Identified four lactate-related genes (LRGs): ZNF662 as risk gene whereas CGNL1, VWCE, and ZFP42 as protective genes, significant for OSCC prognosis. Subgroup analysis based on LRG expression revealed significant PCA differences; subtype 2 had lower LRG scores and better survival prognosis than subtype 1. Variations in stromal, ESTIMATE, and immune scores among subtypes; subtype 1 had the highest stromal score, while both subtype 1 and subtype 2 exhibited high ESTIMATE and immune scores. High-risk scores correlated with increased sensitivity (lower IC50 values) to AZ6102, venetoclax, nutlin_3a, and doramapimod, whereas PD0325901, ERK_6604, and lapatinib showed lower sensitivity (higher IC50 values). Did not report the correlation with immune cell in this study.	N/A

## 6. Resisting Cell Death

Resisting cell death is a key feature of cancer that drives tumor formation, growth, and the ability to withstand therapy. Normal cells are designed to undergo cell death when faced with stress, DNA damage, or uncontrolled replication. This mechanism is crucial for maintaining tissue balance and preventing the buildup of damaged cells that could turn cancerous. In contrast, cancer cells bypass these processes, enabling them to survive in conditions that would normally lead to cell death. This ability fuels their unchecked growth, spread to other locations, and evasion of immune detection. The mechanisms by which cancer cells resist cell death are varied and interconnected. Endoplasmic reticulum stress (ERS) pathways, triggered by the buildup of incorrectly folded proteins, can aid tumor survival by activating adaptive signaling, such as the unfolded protein response (UPR). Cancer stem cells (CSCs), which can both self-renew and differentiate, evade apoptosis and are crucial in recurrence and metastasis. The epithelial–mesenchymal transition (EMT) increases the invasive potential of tumors and concurrently heightens their resistance to apoptotic signals. Chemoresistance, especially to cisplatin, stems from changes in DNA repair mechanisms, cellular transport, and metabolic processes. While cellular senescence initially inhibits tumor growth, it can unexpectedly foster progression through the secretion of pro-inflammatory cytokines and alterations of the TME. This issue carries significant consequences for cancer therapies. The resistance to cell death is a primary reason for treatment failures in radiotherapy, chemotherapy, and immunotherapy. Thus, comprehending the complex mechanisms that allow cancer cells to avoid programmed cell death is essential for creating new therapeutic approaches. By targeting these resistance pathways, there is potential to enhance treatment efficacy, overcome resistance, and ultimately improve outcomes for patients.

### 6.1. Endoplasmic Reticulum Stress-Related Genes (ERS)

The ER is involved in crucial processes like secretion and protein synthesis [153]. A buildup of misfolded proteins because of ER function disruptions sets off the unfolded protein response (UPR) [154]. This adaptive signaling mechanism helps cells maintain homeostasis and regulate stress [155]. Persistent or aberrant activation of the UPR significantly influences neoplastic progression through modulation of cellular proliferation, survival mechanisms, and therapeutic resistance [156]. Therefore, identifying genes associated with ER stress and their link to prognosis can improve treatment plans and outcomes, addressing the aggressive nature of OSCC and the limitations of current therapies. 

Cheng et al. [157] aimed to create a predictive model for OSCC by identifying genes related to prognosis and differentially expressed ERS. Three ERS genes—*IBSP*, *RDM1*, and *RBP4*—were found to be important risk factors for OSCC by examining data from the TCGA and GTEx datasets. To create a prognostic risk model, these genes were chosen using LASSO. It showed strong efficacy in predicting the prognosis of OSCC, with high-risk patients showing increasing IC50 values for docetaxel, gefitinib, and erlotinib, indicating higher treatment resistance compared to low-risk patients who exhibited increased sensitivity to these drugs. Strong relationships between the risk score and other variables, such as immune cell infiltration, stem cell properties, and TMB, were also revealed by multi-omics analysis. More infiltration of monocytes, resting mast cells, and eosinophils was specifically seen in high-risk patients, indicating a unique immunological landscape linked to elevated risk. In Cox regression analysis, the risk score also showed up as an independent prognostic factor, highlighting its potential as a trustworthy instrument for patient outcome prediction. The model’s exceptional predictive ability and its association with immune microenvironment changes and medication sensitivity highlight its clinical significance and potential to inform individualized treatment plans for OSCC.

### 6.2. Cancer Stem Cells (CSCs)

A crucial subpopulation found in malignancies, CSCs are distinguished by stem cell-like properties such as asymmetric division and self-renewal that produce a variety of cancer cell types [158]. Compared to non-CSC subpopulations, CSC is more robust in unfavorable microenvironments, highly invasive, and noticeably resistant to apoptosis. The characteristics of CSCs, including resistance to treatment, dormancy, and evasion of cell death, highlight their significance in the advancement and reappearance of tumors. Consequently, concentrating on CSCs is one possible therapeutic approach. Nevertheless, there are currently insufficient and trustworthy biomarkers for identifying and classifying CSCs in OSCC. Finding new CSC markers that correlate with changes in OSCC cancer growth is crucial to developing successful treatments.

In 2024, Shi et al. and colleagues identified a six-gene cancer stem cell-related prognostic signature (6-GPS) for OSCC using data from single-cell RNA sequencing [159]. Through LASSO Cox regression analysis, the genes *PGK1*, *POLR1D*, *PTGR1*, *P4HA1*, *ADM*, and *RPL35A* were selected from a total of 1344 cancer stem cell-related genes to form the 6-GPS. In the OSCC samples, PTGR1 mRNA expression was significantly reduced, while levels of *PGK1*, *RPL35A*, *POLR1D*, *P4HA1*, and *ADM* were elevated. Immunohistochemistry confirmed these results, showing that OSCC had higher protein abundances of PGK1, POLR1D, P4HA1, and ADM compared to normal mucosa. Functional tests indicated that the suppression of POLR1D and ADM led to decreased cell proliferation and reduced G1 phase cell cycle arrest. Additionally, it prevented the synthesis of important signaling molecules such as CD133, HIF1α, JAK1, and CCND1. Numerous methods, including nomograms, calibration plots, and ROC curves, were used to confirm the 6-GPS’s prognostic accuracy. The ICGC dataset provided more evidence in favor of this. The findings underscore the critical roles these genes play in tumor growth and metastasis, demonstrating the utility of the 6-GPS as a diagnostic and prognostic tool that provides information on the course of OSCC and potential targets for treatment.

### 6.3. Epithelial–Mesenchymal Transition (EMT)

An epithelial cell that has undergone polarization can go through a reversible dynamic mechanism called epithelial–mesenchymal transition (EMT) to give rise to a mesenchymal cell phenotype. This phenotype includes increased invasiveness and migration ability, increased resistance to apoptosis, and a noticeably increased production of ECM components [160]. Therefore, identifying gene signatures associated with EMT in OSCC that serve as potential predictors of prognosis and offer valuable insights into the TIME and underlying molecular mechanisms of OSCC could aid in developing targeted treatment approaches. 

Ai et al. [161] aimed to construct and validate an EMT gene profile to predict the prognosis of OSCC. Their initial study identified eleven prognostically significant EMT-related genes: *SFRP1*, *GAS1*, *COL5A3*, *TNFRSF11B*, *VEGFA*, *DKK1*, *FMOD*, *GPX7*, *ANPEP*, *PLOD2*, and *AREG*, through univariate Cox regression analysis of TCGA data. This list was narrowed down to nine genes (*VEGFA*, *AREG*, *DKK1*, *PLOD2*, *SFRP1*, *TNFRSF11B*, *GPX7*, *COL5A3*, and *GAS1*) via LASSO regression, which made it possible to create a risk score formula that successfully divided patients into high-risk and low-risk categories. When compared to low-risk patients, high-risk patients had far worse survival outcomes, and the risk score showed better predictive value than conventional clinical criteria. These results were validated externally using the GSE41613 cohort, where higher-risk patients once again had lower survival periods. Using the ESTIMATE method and CIBERSORT, a study of the immune microenvironment revealed that high-risk OSCC samples exhibited higher stromal and immune scores but lower tumor purity. It was found that activated memory CD4 T cells, active mast cells, aDC, resting NK cells, and eosinophils were significantly more prevalent in high-risk samples compared to naïve B cells, Tfh cells, Tregs, Tgd cells, and resting mast cells. This altered immune profile in high-risk patients indicates a more complex and potentially more immunologically active TME. GSEA showed upregulation of DNA repair pathways, cell cycle, and basal transcription factors in high-risk samples, while pathways like VEGF and MAPK signaling were activated in low-risk samples. Somatic mutation analyses indicated that high-risk samples had a higher frequency of mutations, particularly in genes such as *FAT1*, *TP53*, and *CDKN2A*. Expression analysis further confirmed the overexpression of COL5A3, GAS1, GPX7, AREG, and PLOD2 in OSCC tissues, while SFRP1 was notably downregulated. According to the study’s findings, the EMT gene signature is a strong predictor of outcome and offers insightful information about the immunological environment and underlying molecular mechanisms of OSCC, information that may help develop individualized treatment plans.

### 6.4. Cisplatin Resistance 

Cisplatin is an alkylating agent that breaks down DNA by forming cisplatin-DNA adducts that result in cell cycle arrest and programmed cell death [162]. Drug transport, DNA damage and repair, and programmed cell death are among the processes that contribute to resistance to platinum-based medicines in OSCC [163]. Lately, the TME and epigenetic processes have also come to light as significant components of chemoresistance mechanisms. Finding cisplatin-resistance biomarkers is essential to forecasting treatment outcomes and prognosis because cisplatin resistance has been linked to a worse prognosis for patients with OSCC [164].

In the work of Lu et al. [165], a predictive model for OSCC was developed using seven cisplatin-resistance-related genes: *STC2*, *TBC1D2*, *ADM*, *NDRG1*, *OLR1*, *PDGFA*, and *ANO1*. Information on gene expression linked to cisplatin resistance and OSCC was obtained from the TCGA and GEO databases. By comparing the overlapping DEGs between the cisplatin-resistant and parental samples, as well as between the tumor and control groups, cisplatin-resistant genes were found. Prognosis-related genes were further refined by univariate Cox regression and LASSO regression experiments, resulting in the development of a prognostic risk-score (RS) model. GSEA was used to identify significant differences in pathways between the high-risk and low-risk groups. These modifications included those related to EMT, hypoxia, and oxidative phosphorylation pathways. The CIBERSORT-assisted immune landscape analysis showed significant differences in immune cell infiltration: more CD8 T cells, plasma cells, naïve B cells, active memory CD4 T cells, Tfh cells, Tregs, M2 macrophages, and resting mast cells were seen in the low-risk group. However, this group had lower levels of M0 macrophages, resting NK cells, naïve CD4 T cells, and active mast cells. Immunological checkpoint markers were also expressed differently in the two risk groups: *PDCD1*, *CD96*, *CTLA4*, *TIGIT*, and *LAG3* were overexpressed in the low-risk group, whereas *PVR* was underexpressed. Additionally, the predicted IC50 values of 57 medicines showed significant differences between risk categories. According to these results, immune cell profiles and genes that have been found may play a critical role in chemoresistance and prognosis prediction in OSCC, providing information for the development of targeted treatments.

### 6.5. Cellular Senescence

Cellular senescence represents a complex, programmed stress response characterized by irreversible cell cycle arrest and distinctive molecular alterations in response to diverse intrinsic and extrinsic stimuli [166]. Cellular senescence is a term used to describe a permanent cell cycle arrest that can affect many different biological processes, both normal and pathological. Senescent cells have a variety of roles in the development of cancer. At times, they secrete substances that fuel inflammation and metastasis, which can impede tumor growth through mechanisms such as cell cycle arrest and apoptosis. In addition, chemokines, cytokines, and growth factors secreted by senescent malignant cells have a significant impact on the remodeling of the TME and lead to dysregulated immune cell infiltration [167]. Studies reveal that cellular senescence may contribute to tumor growth and is associated with unfavorable prognoses in various cancers [168]. Therefore, identifying targetable gene signatures to reverse the effects of cellular senescence in OSCC is critical. However, cellular senescence-associated genes solely to OSCC remain largely undetermined. 

Lately, Wang et al. [169] study created a strong prognostic model for OSCC by finding and examining seven new senescence-related genes (SRGs) that are connected to the prognosis and response to treatment for OSCC: *CDK1*, *G6PD*, *IL1A*, *MAD2L1*, *PDCD10*, *PTTG1*, and *VEGFA*. The role of *CDK1* in OSCC was confirmed through functional validation using siRNA-mediated CDK1 knockdown in vitro, which resulted in a senescence phenotype marked by elevated production of *p21* and *SA-β-Gal* and decreased cell proliferation. According to the results, patients with high SRG scores were far more sensitive to chemotherapeutic drugs, including docetaxel, cisplatin, and 5-fluorouracil, as seen by their lower IC50 values as compared to those with low SRG scores. This implies that the SRG signature correlates with chemotherapy sensitivity in addition to acting as a prognostic marker. Patients exhibiting low SRG scores demonstrated elevated stromal and immune scores within the TME, suggesting an enhanced immunological landscape. High SRG scores, on the other hand, were linked to decreased levels of B cells, B cell memory, CD4 Tem cells, CD8 Tcm cells, and CD8 Tem cells. This decrease in immune cell populations adds to a more immunosuppressive TME and suggests a possible compromise in anti-cancer immune responses. The SRG signature and these immune cell types have a negative correlation, which emphasizes how higher SRG scores may be associated with compromised immune surveillance, which could lead to tumor development and affect the effectiveness of treatment. Overall, these results validate the effectiveness of the SRG-derived nomogram and signature as predictive instruments for OSCC prognosis and chemotherapeutic response and highlight their significance in clarifying the intricate interactions among cellular senescence, immunological environment, and treatment effects.

**Table 4 biomedicines-13-00134-t004:** Overview of prognostic gene signatures that are associated with resisting cell death pathways.

Authors	Findings	FDA Approved Drugs (Launch/Phase)
Ai et al. [161]	Identified a prognostic nine-epithelial–mesenchymal transition (EMT) gene signature includes VEGFA, AREG, DKK1 PLOD2 as risk genes whereas SFRP1, TNFRSF11B, GPX7, COL5A3, GAS1 as protective genes. High-risk group exhibited higher number of activated memory CD4 T cell, resting NK, aDC and mast cell, and lower number of naïve B cell, Tfh, Treg, Tgd cells and resting mast cells.	VEGFA (pidolic-acid, trometamol, vandetanib; DKK1 (KY02111); SFRP1 (WAY-316606); GPX7 (Glutathione)
Cheng et al. [157]	Identified prognosis-related and differentially expressed endoplasmic reticulum stress-related genes (ERS) in OSCC. Three ERS genes IBSP, RDM1, RBP4 risk genes were identified. These ERS gene signatures was associated with higher number of monocytes, resting mast cells and eosinophils. High-risk group of patients had higher IC50 values for docetaxel, gefitinib, and erlotinib, suggesting that low-risk patients may be more sensitive to these drugs.	RBP4 (A-1120)
Shi et al. [159]	Identified a 6-gene cancer stem cell-related prognostic signature (6-GPS). The 6-GPS are ADM, POLR1D, PTGR1, RPL35A, PGK1, and P4HA1 (risk genes) The mRNA expression of the 6-GPS was analyzed. OSCC sample exhibited lower expression of PTGR1 while higher expression of ADM, RPL35A, PGK1, POLR1D and P4HA1 (*p* < 0.05). Immunohistochemistry results revel protein abundance of 6-GPS. Notably, higher protein level of PTGR1, PGK1, POLR1D and P4HA1 in OSCC samples compared to normal skin mucosa. Knockdown of ADM and POLR1D reduced the cell proliferation, induced cell cycle arrest, stem cell marker CD133, and phosphorylation of JAK1, HIF1α and CCND1 expression as compared to control in in vitro of OSCC preclinical models. Did not report the correlation with immune cell in this study.	PTGR1 (naringenininc-acid); P4HA1 (L-proline, succinic-acid).
Xu et al. [114]	Identified 14 lncRNA-based signature as a potential prognostic biomarker in OSCC. Based on 14 lncRNAs (AFAP-AS1, ALMS1-IT1, HLA-F-AS1, LINC-PINT, LINC00958, NPSR1-AS1, PRKG1-AS1, and WDFT3-AS2 as risk gene and KANSL1-AS1, LINC00567, LINC00689, LINC00877, LINC01191, LINC01281 as protective gene), patients were divided into low- and high-risk subgroups with different survival times. The low-risk group showed significantly better prognosis compared to high-risk. Low-risk group was associated with higher number of aDC, eosinophils, macrophage, mast cell, monocyte, MDSC, NK, neutrophils, pDC, activated B cells, activated CD8 T cells, Tcm central memory for CD4 and CD8, Tem CD8 T cell, Tgd, immature B cell, Treg, Tfh, Th1 and Th17 cells compared high-risk groups. Distinct immunocytes infiltration levels between low and high-risk groups demonstrated negative correlation for monocytes, activated B cells, macrophages, activated CD8 T cells, Tfh, and MDSCs in correlation analysis.	N/A
Lu et al. [165]	Identified seven cisplatin-resistance-related prognostic signatures: STC2, TBC1D2, ADM, NDRG1, OLR1, PDGFA, and ANO1 (risk genes). KM analysis indicated significantly better prognosis in the low-risk group compared to the high-risk group. Low-risk group was associated with higher number of naïve B cells, plasma cells, CD8 T cells, activated memory CD4 T, Tfh, Treg, M2 macrophage, resting mast cells and lower number of naïve CD4 T cells, resting memory CD4 T cell and resting NK cells, M0 macrophage and activated mast cells. Five immune checkpoint markers (PDCD1, CD96, CTLA4, TIGIT, and LAG3) were found overexpressed whereas PVR was found under expressed in low-risk groups when compare to high-risk groups.	ANO1 (fluoxetine, niflumic-acid, tannic-acid)
Wang et al. [169]	Identified seven novel senescence-related genes (SRGs) in OSCC prognosis: CDK1, G6PD, IL1A, MAD2L1, PDCD10, PTTG1, and VEGFA (risk genes). IC50 values of chemotherapy drugs indicated higher sensitivity to 5-fluorouracil, cisplatin, and docetaxel in patients with high SRG scores (with lower IC50 compare with low-risk groups). Low-score subgroup exhibited higher immune and stromal scores in the TME compared to the high-score subgroup. SRG signature scores showed negative correlations with B cells, B cell memory, CD4 Tcm cells, CD4 Tem cells, CD8 Tcm cells, and CD8 Tem cells. Knockdown of CDK1 induced expression of p21 and SA-β-Gal expression as well as inhibit cell proliferation in in vitro preclinical model of OSCC.	CDK1 (adenosine-triphosphate, alvocidib, AT-7519, dinaciclib, indirubin, PHA-793887); G6PD (dehydroepiandrosterone, RRx-001); VEGFA (pidolic-acid, trometamol, vandetanib)

## 7. Immune Response and Tumor Microenvironment

The concept that the immune system has a dual role in the development of cancer has been bolstered by growing experimental data in recent decades [170]. It is necessary for the identification and destruction of tumor cells, but it can also encourage the selection of tumor variations that are less immunogenic, which makes it possible for cancer cells to elude immune monitoring. The cancer immunoediting hypothesis, which acknowledges the immune system’s multiple involvement in tumor promotion and suppression during the development of cancer, is based on this dual action [171,172]. We examine the role immune cells play in the development of OSCC in this article. While the immune system plays a crucial role in eliminating neoplastic cells, manipulation of the TME can lead to the emergence of malignant cells with immune-evasive properties. This dynamic interplay between the TME and immune components can result in the selection of cancer cell populations capable of escaping immunosurveillance, potentially promoting tumor progression and therapeutic resistance.

One potentially effective treatment for OSCC is to target the TIME [173]. The functions that immune cells and their interactions within the TME play in either promoting or preventing tumor growth, invasion, and metastasis are now well known. Making use of these cells and elements offers a compelling plan to strengthen the immune system’s defense against cancerous cells. The functions of the gene signature associated with prognosis to different immune cells within the TIME of OSCC, such as TAMs, MDSCs, Tregs, CD8 T lymphocytes, and NK cells, and their contributions to either promoting or impeding tumor progression, are investigated in this review study [174]. Despite substantial efforts to investigate the tumor-related host response in OSCC, the fundamental principles, mechanisms, and molecules involved have not yet been fully integrated into clinical practice. For example, the eighth Edition of the American Joint Committee on Cancer (AJCC) introduced new prognostic criteria, such as depth of invasion (DOI) and extranodal extension (ENE), for assessing tumor (T) and nodal (N) parameters [175]. However, immune-related characteristics have not been included in these criteria. Recent guidelines from the International Immuno-oncology Working Group, which highlight the importance of analyzing TILs, represent a significant step towards potential clinical application [176]. Identifying specific immune cell subpopulations through molecular biomarkers seems to be a promising strategy. As research delves deeper into the complex interactions within the TME, recognizing key immune-related genes and markers associated with CAFs has become crucial for understanding the mechanisms driving OSCC and for developing innovative therapeutic strategies. These efforts allow for clarification of potential molecular signatures to better stratify patients, predict outcomes, and tailor treatment approaches, providing new hope in the battle against OSCC.

### 7.1. Immune Gene Signatures and Prognosis

The study by Zhu et al. [177] aimed to identify immune-related genes with differential expression in OSCC tumors compared to normal tissues. They uncovered 16 genes associated with survival through univariate Cox regression analysis, suggesting these could serve as prognostic biomarkers. A nine-gene risk signature was established, including *APOD*, *OLR1*, *STC2*, *DKK1*, *TNFRSF19*, *TNFRSF4*, *DEFB1*, *CTLA4*, and *CTSG*. In high-risk patients, *APOD*, *OLR1*, *STC2*, and *DKK1* were found to be overexpressed, with *APOD* showing tumor-suppressive effects in certain cancers, while *OLR1*, *STC2*, and DKK1 are known to promote tumor growth and immune suppression. Conversely, the protective genes identified were *TNFRSF19*, *TNFRSF4*, *DEFB1*, *CTLA4*, and *CTSG*. Increased expression of *TNFRSF19* has been linked to poor prognosis in some cancers, whereas *TNFRSF4* (OX40) enhances CD8 T cell infiltration [178,179]. *DEFB1* inhibits tumor invasion and migration in OSCC, and *CTLA4*, a negative regulator of T cell activation, has shown efficacy in boosting anti-tumor immunity when targeted with CTLA4 inhibitors [180,181]. Apart from that, an analysis of TCGA HNSCC RNA-seq data demonstrated that low expression of *CTLA4* was associated with poor prognosis. *CTSG* has emerged as a significant immune-related biomarker in OSCC due to its role in reducing tumor cell proliferation and invasion [182]. The nine-gene signature effectively categorized OSCC patients into high-risk and low-risk groups, revealing significant differences in survival rates. Higher proliferation and wound healing scores correlated with poorer outcomes in the high-risk group. Immune infiltration analysis indicated that the low-risk group had elevated immunological scores, marked by increased lymphocyte and macrophage infiltration, along with a robust IFNG response. This group also showed higher levels of anti-tumor immune cells, such as activated CD4 and CD8 T cells and NK cells. In contrast, the high-risk group exhibited a “cold tumor” phenotype with limited immune cell presence, while the low-risk group displayed a “hot tumor” phenotype characterized by a complex immune landscape featuring both anti-tumor and immunosuppressive cells, like Tregs, macrophages, and MDSCs. This dual immune infiltration likely contributes to the improved prognosis observed in the low-risk group, underscoring the intricate TME interactions.

Chen et al. [183] developed a prognostic model based on immune-related genes in OSCC, identifying *CTSG*, *TNFRSF4*, *IGLV1-44*, *STC2*, and *CCL22* as significant prognostic markers. *CTSG* plays a role in neutrophil activation and immune response, while *TNFRSF4* (OX40) is a favorable target for the immunotherapy [184]. *IGLV1-44* is involved in B cell differentiation, and STC2 promotes tumor development across multiple cancers [185,186]. *CCL22* recruits Tregs to the TME, suppressing the anti-tumor immunity [187]. Li et al. [188] performed a comprehensive analysis of immune cells and immune-related genes in OSCC, linking stromal and immune components of the TME, as measured by ESTIMATE scores, to longer OS. Differential expression analysis between high- and low-score groups identified 593 DEGs enriched in pathways associated with T cell receptor signaling and immunoglobulin complexes. LASSO and random forest models identified 11 key immune-related genes, including *CCL22*, *FLT3*, and IL10, as significant predictors of OSCC patient survival.

Lv et al. [189] explored the immune landscape of OSCC, dividing patients into Immune_High, Immune_Medium, and Immune_Low groups based on immune cell infiltration scores. A total of 854 immune-related genes were identified between the Immune_High and Immune_Low groups, with enrichment in processes such as leukocyte activation, adaptive immune responses, and lymphocyte activation, which are critical in modulating OSCC malignancy. In the context of hypoxia, 193 overlapping immune-related genes were analyzed, leading to the identification of eight mRNA signatures associated with OS: *FAM122C*, *RNF157*, *RANBP17*, *SOWAHA*, *KIAA1211*, *RIPPLY2*, *INSL3*, and *DNAH1*. These genes are implicated in various cancers. For instance, *KIAA1211* is overexpressed in non-small cell lung cancer, and its knockdown inhibits cell proliferation while inducing apoptosis [190]. *INSL3* is known to promote tumor growth and angiogenesis in several cancers, making it a potential OSCC prognostic marker [191].

### 7.2. Arecoline-Associated Fibrosis

Li et al. [192] emphasized the role of CAFs in OSCC, which are non-immune components of the TME that promote tumor progression. CAFs drive cancer through immunosuppression, metabolic shifts, proliferation, angiogenesis, invasion, and therapy resistance [193,194]. Clinically, OSCC patients with elevated CAF expression had higher rates of lymph node metastasis and significantly worse survival [195,196]. Despite the significance of CAFs, therapies targeting these cells have had limited clinical success, largely due to gaps in understanding their molecular and functional characteristics. Previous research by the authors highlighted that arecoline, a known risk factor for OSCC, increases phosphodiesterase-4A (PDE4A) activity in TGFβ activated buccal mucosal fibroblasts, which contributes to the development of oral submucous fibrosis (OSF), a precancerous condition [197]. This study further explored the link between arecoline-induced OSCC progression and CAF-related gene expression, identifying key genes such as *COL1A2*, *PLAU*, and *CCL11* that were strongly associated with CAF activity. Wu et al. [198] investigated the relationship between OSCC and periodontal disease, noting that fibroblast activity is key in tumor progression, wound healing, and immune evasion. They applied ssGSEA to assess CAF scores in OSCC patients and found correlations between CAF expression and immune-related genes, particularly HLA and immune checkpoint genes. Six key genes—*ACTN2*, *AQP1*, *IL10*, *PLAU*, *SLC2A3*, and *TIMP4*—were associated with both periodontal disease and OSCC, suggesting their potential as biomarkers for OSCC progression.

The exploration of immune-related genes, CAFs, and the TME has provided significant impacts and insights in relation to the underlying pathways of OSCC progression and patient prognosis. As research continues to unravel the complexities of the immune landscape, novel prognostic biomarkers and therapeutic targets are emerging, offering the potential for more personalized and effective treatment strategies. The identification of specific immune signatures and CAF-related genes not only aids in risk stratification but also underscores the importance of targeting the TME in future therapeutic approaches. By leveraging these molecular insights, the field is poised to improve outcomes for OSCC patients, bringing the prospect of better survival rates and more targeted treatment options closer to reality.

**Table 5 biomedicines-13-00134-t005:** Overview of prognostic gene signatures that are associated with immune response and TME pathways.

Authors	Findings	FDA Approved Drugs (Launch/Phase)
Zhu et al. [177]	Identified the candidate immune-related genes (IRGs) that related to the prognosis and immune landscape of OSCC. Nine immune-related risk genes were established (APOD, OLR1, STC2, DKK1, TNFRSF19, TNFRSF4, DEFB1, CTLA4 and CTSG). Among these IRGs, four genes (APOD, OLR1, STC2 and DKK1) were regarded as risk genes, while the remaining genes (TNFRSF19, TNFRSF4, DEFB1, CTLA4 and CTSG) were identified as protective genes. Expression of CTSG, CTLA4, TNFRSF4, APOD and OLR1 was positively correlated with the enrichment score of most immune cells but STC2 was negatively correlated with the enrichment score of most immune cells. Low-risk group was associated with high anti-tumor immune cells (activated CD4 T cells, activated CD8 T cells, CD4 T central memory (Tcm) cell, NK cells, CD8 T effector memory (Tem) cell, Th1 cell, Th17 cell, NK killer T cell, and high Treg cells, macrophages, myeloid-derived suppressor cells (MDSCs).	N/A
Li et al. [192]	Identified arecoline-associated fibrosis-related genes (AFOC) on the inhibition of cuproptosis and proliferation of cancer-associated fibroblasts (CAFs) in OSCC. A total of 13 genes related to arecoline were identified and all of which were upregulated. Among those, AFOC-differential gene expression (DEGs) differentially expressed genes were identified (PLAU, IL1A, SPP1, CCl11, TER, COL1A2). Correlation of AFOC-DEGs and cuproptosis was also studied (PLAU and IL1A reported highest) Immune infiltration analysis showed there is an association between AFOC-DEGs (COLIA2, PLA, CCL11) and CAFs but the immune cell population was not reported in this study.	PLAU (amiloride, mexiletine);
Lv et al. [189]	Identified hypoxia-immune—based gene signature related to OSCC prognosis. Eight signature mRNAs were associated with OS were identified (FAM122C, RNF157, RANBP17, SOWAHA, KIAA1211, RIPPLY2, INSL3, and DNAH1) and regarded as protective genes. All eight signatures genes were highly expressed in the low-risk group than high-risk group. Low-risk group is associated with higher degree of infiltration of CD8 T cells, plasma cells, Tfh cells, Treg and naive B cells. High-risk group is associated with higher aDCs, activated mast cells and neutrophils.	N/A
Chen et al. [183]	Identified immune-related gene signature that associated with OSCC prognosis linked to the immune response. A total of 18 immune related DEGs were strongly linked with OS among OSCC patients were identified in the immunity clusters and ImmPort databases. Among those, five immune-related genes signatures were identified (CTSG, TNFRSF4, IGLVQ-44, STC2, CCL22) from the immune infiltration cluster by LASSO regression analysis. CTSG, TNFRSF4, IGLVQ-44 and CCL22 are regarded as protective genes and STC2 is regarded as risk gene. The correlation of these signature genes and immune cell infiltration was assessed using CIBERSORT. CTSG was associated with low number of resting mast cells and naïve B cells. CCL22 was associated with high number of neutrophils, eosinophils, activated mast cells, resting mast cells, aDCs, M1 macrophages and low number of resting NK, Tregs, Tfh, activated memory CD4 T cells and naive B cells. IGLV1-44 was associated with high number of neutrophils, activated mast cells, resting mast cells, M0 macrophages and low number of activated NK cells, Tfh cells, activated memory CD4 T cells, naïve CD4, B cells, and CD8 T cells, plasma cells, naïve and memory B cells. TNFRSF4 was associated with high number of eosinophils, activated mast cells, resting mast cells, M0 and M2 macrophages and low number of Tregs, Tfh cells, CD8 T cells, naïve B cells. STC2 was associated with high number of eosinophils, activated and resting mast cells, resting DCs, M1 and M0 macrophages and low number of activated NK cells, resting NK cells, Tfh cells and CD8 T cells. Ectopic expression of both CTSG and TNFRSF4 inhibit cell proliferation, migration and invasion ability in in vitro of OSCC preclinical models as compared to control.	CTSG (aloxistatin, delanzomib);
Wu et al. [198]	Identified association between CAFs in OSCC and explores the potential correlation between OSCC and periodontal disease. Six CAFs (ACTN2, AQP1, IL10, PLAU, SLC2A3, and TIMP4) were identified as a key prognostic factor in the OSCC. ACTN2, PLAU, SLC2A3, and TIMP4 high expression linked to poor OS (risk gene), where AQOP1 and IL10 correlated with better OS in OSCC (protective gene). OSCC cohort was divided into immune-low and immune-high groups. In correlation analysis, the expression of immune checkpoint-related genes was significantly associated with the immune-scores (score was highly associated with high expression of HLA-related genes, such as, HLA-A, HLA-B, HLA-C, HLA-DMA). Multiple immune-related cells (NK, T cell, M1, M0, M2 macrophages, CAFs and mast cell) were associated with CAFs-based periodontal disease.	AQP1 (acetazolamide); IL10 (JTE-607); PLAU (amiloride, BC-11, mexiletine, 4-chlorophenylguanidine); SLC2A3 (2-deoxyglucose).
Li et al. [188]	Identified 11 immune-related gene signatures namely AC103563.1, CCL22, FLT3, IGLV4.60, IL10, LINC00861, MS4A2 (protective gene), and GALR2, LINC01508, IGKV1D.8, and IGLV1.36 as risk gene in OSCC prognosis. GALR2, LINC01508, IGKV1D.8, and IGLV1.36 exhibited high expression in the high-risk group and were negatively correlated with OS.	FLT3 (ceritinib, gilteritinib, midostaurin); IL10 (JTE-607)

## 8. The Prognostic Impact of Tregs in OSCC: Protective Role and Clinical Implications

### 8.1. Protective Role of Tregs in OSCC

Tregs within the TME are known to have a significant impact on cancer progression [199]. These Tregs exhibit immunosuppressive capabilities in the TME, thereby fostering tumor survival and progression through the modulation of immune responses. Numerous studies have highlighted a correlation between elevated Treg numbers and unfavorable clinical outcomes in the HNSCC [200,201,202]. Our study, depicted in Table 1, revealed that 13 studies [54,55,67,77,80,88,93,114,129,148,165,177,189] showed a positive connection between increased Treg levels and protective gene signatures that are linked to positive outcomes in OSCC. These findings agree with studies [203,204,205,206,207], which suggest that elevated FoxP3 Treg infiltration corresponds to improved OS rates in HNSCC patients. Studies conducted over the past decade by Shang et al. [208] also demonstrated the positive impact of FoxP3 Tregs on survival outcomes in colorectal, head and neck, and esophageal cancers, with outcomes varying based on cancer subtype and stage.

Mandal et al. [26] utilized transcriptomic data from TCGA to assess the HNSCC immune landscape, finding that HPV-positive cases had significant Treg infiltration, followed closely by HPV-negative cases. They also discovered that high levels of CD8 T cells and Tregs were associated with enhanced survival rates, even after adjustment of HPV status as the confounding factor in Cox regression analyses. Interestingly, the relationship between Treg levels and other T cell populations like CD8 cytotoxic T lymphocytes was deemed particularly strong. Nonetheless, once these other immune cells were accounted for, Treg levels themselves were not deemed independent prognostic indicators, hinting that Tregs may have a more reactive role rather than a direct influence on prognosis.

Bron et al. [205] further illustrated that FoxP3 cell numbers were notably higher in tumors of the oral cavity and oropharynx compared to those in the hypopharynx and larynx. Higher counts of these cells in both intraepithelial and stromal regions correlated with metastatic lymph nodes negative, indicating a favorable prognosis. The origin of these FoxP3 Tregs may hold significance in terms of either aiding in tumor control or promoting tumor progression by inhibiting inflammatory processes. In HNSCC patients, Badoual et al. [203] found that FoxP3 CD4 Tregs infiltrating tumors were linked to improved locoregional tumor control. Their analysis identified T stage and Treg infiltration as the key prognostic factors for locoregional control. Additionally, Strauss et al. [209] observed higher frequencies of FoxP3 CD4 Tregs in patients who showed no signs of disease following oncologic treatment compared to those with active disease.

Interestingly, our study uncovered a curious paradox: both low [54,55,67,77,80,88,93,114,129,148,165,177,189] and high [143,183] Treg levels were associated with poor prognosis in OSCC. This finding might be explained by the study of Erdman and Poutahidis [210], which suggested that Tregs from uninfected mice failed to suppress cancer development due to lower IL10 expression compared to Tregs from H. hepaticus infected mice. This shift in Treg phenotype could be influencing tumor progression, especially in inflammatory environments like those found in the OSCC/HNSCC region [211,212]. In other words, the constant exposure of the oral mucosa to resident microbiota creates a scenario akin to that in colon cancers, where high Treg density is linked to positive outcomes. The interaction between microbiota and immune cells in the oral cavity has the potential to alter Treg function, influencing tumor progression [213]. Additionally, research by Yamazaki et al. [214] showcased the protective role of FoxP3 Tregs in local oral infections, hinting at a potential counteraction of pro-tumorigenic effects caused by inflammation through the infiltration of Tregs induced by the septic environment of the oral mucosa, thereby promoting anti-inflammatory responses and tissue homeostasis. This process plays a crucial role in preventing the development of OSCC by encouraging anti-inflammatory reactions and preserving tissue equilibrium.

### 8.2. Clinical Applications of Tregs in OSCC Management

The dual characteristics of Tregs in OSCC create both opportunities and challenges for clinical applications. Elevated levels of Tregs have demonstrated the potential to enhance locoregional tumor control and overall survival, especially when they are balanced with the activity of cytotoxic T cells. This underscores the necessity for therapies that modify Treg function instead of merely reducing Tregs. Current clinical approaches utilizing Tregs include immune checkpoint inhibitors like CTLA4 inhibitors, which boost anti-tumor immunity while maintaining Treg–mediated tissue balance. Research conducted by Badoual et al. [203] and Mandal et al. [26] emphasizes the significance of sustaining a proper Treg–CD8 T cell balance for successful therapies. Moreover, gaining insights into how the microbiota influences Treg phenotypes may lead to microbiome-targeted therapies that can alter the immune landscape in OSCC. Innovative therapies that focus on Tregs also encompass adoptive T cell therapies and cytokine-based strategies designed to adjust the tumor-promoting or suppressive roles of Tregs. For example, strategies that encourage the activity of IL10-expressing Tregs may amplify anti-inflammatory effects and mitigate OSCC advancement in highly inflammatory settings. In summary, incorporating Treg-related biomarkers into diagnostic and prognostic frameworks could enhance the stratification of patients and the personalization of treatments in OSCC, aligning therapeutic approaches with the complex roles of Tregs in the TME.

## 9. Future Directions for FDA-Approved Drugs in OSCC 

It is important to enhance the range of potential therapy, particularly considering the promising efficacy of various FDA-approved medications such as L-aspartic acid, L-cysteine, L-glutamic acid, MC-1 (targeting *GOT1*), bimatoprost (targeting AKR1C3), ephedrine-(racemic) (targeting *ATF6*), PGL5001 or RGB-286638 (targeting *MAPK9*), ethoxzolamide or indisulam (targeting *CAIX*), pidolic acid and trometamol (targeting *VEGFA*), AZD4282 (targeting *SHMT2*), C11-Acetate and polyiosine (targeting *HPRT1*), cladribine (targeting *POLD2*), adenosine-phosphate, ribavirin (targeting *ADK*), glutathione (targeting *MGST1*), vidarabine (targeting *ADA*), Naringeninic acid (targeting *PTGR1*), oseltamivir-phosphate (targeting *Neu1*), A-1120 (targeting *RBP4*), acetazolamide (targeting *AQP1*), amiloride, mexiletine, 4-Chlorophenylguanidine (targeting *PLAU*), tannic acid (targeting *ANO1*), and JTE-607 (targeting *IL10*) which have been identified in Table 1 as potential novel inhibitors capable of overcoming treatment resistance in OSCC/HNSCC. For example, RBP4, a transport protein for vitamin A, was identified as a risk gene in the Cheng et al. [157] study, and its inhibitor A-1120 was subsequently discovered. Despite being identified as a risk gene, *RBP4* was found to be downregulated in the same study. *RBP4* has been involved as a moderator contributing to insulin resistance and metabolic disorders [215]. Wang et al. [216] have demonstrated that overexpression of *RBP4* promotes tumor cell migration in preclinical models of ovarian cancer. Additionally, Karunanithi et al. [217] have shown that *RBP4* plays a role in promoting tumorigenesis by regulating self-renewal in colon cancer. The agent A-1120 shares similarities with metformin in addressing insulin resistance, with metformin having demonstrated anti-cancer properties in the treatment of HNSCC/OSCC [218]. Another identified drug, oseltamivir-phosphate, traditionally used to treat influenza infections, was found to target the *Neu1* gene, the mechanism of which in oral tumorigenesis remains unstudied [219]. *Neu3* has been identified as a critical molecule that governs the EGFR signaling pathway and MMP expression preceding EGFR by modulating gangliosides, a signaling process closely linked to lymph node metastasis in the HNSCC [220]. Chuang and colleagues have carried out a large-scale population-based cohort study on a national level and have revealed that the use of oseltamivir may have a protective effect in lowering the risk of various types of cancer, including OSCC [221]. These results indicate a potential association between influenza infection and squamous cell carcinoma, offering a logical justification for the reduced rates of oral and esophageal SCC in individuals using oseltamivir. 

### 9.1. Targeting Hypoxia and Tumor Metabolism 

Hypoxia is widely acknowledged as a significant factor driving tumor advancement and adaptation, leading to more aggressive tumor characteristics by increasing resistance to chemotherapy and limiting the effectiveness of immune checkpoint inhibitors (ICIs) [222,223]. This is due to the hypoxic environment in the TME, allowing resistant cancer cells to survive, evade immune responses, and decrease the efficacy of treatments such as chemotherapy and immunotherapy. In order to address these obstacles, Table 1 has identified acetazolamide as a carbonic anhydrase inhibitor used in the treatment of glaucoma and altitude sickness. Acetazolamide has been found to block AQP1, a channel protein known to play a role in the hypoxia pathway by facilitating the movement of water across cell membranes, a process closely associated with hypoxia in various tumors [224,225]. Specifically, potential inhibitors for *CAIX* have been identified, including coumarin, curcumin, ellagic acid, ferulic acid, saccharin, mafendie (as natural compounds), hydrochlorothiazide and hydroflumethiazide (known as thiazide diuretics approved for hypertension), and zonisamide used as an anti-convulsant agent in the treatment of epilepsy, which have not previously been reported as agents capable of inhibiting CAIX activity as compared to para-toluenesulfonamide and U-104 which have been documented as CAIX inhibitors [226,227]. VEGFA is a promising target in hypoxia, as it is crucial in stimulating angiogenesis, enabling tumors to adjust to low oxygen levels by forming new blood vessels [228]. Our research has identified pidolic acid and trometamol as potential inhibitors of VEGFA. By targeting this protein, we have explored new therapeutic approaches that focus on tumor metabolism and pathways driven by hypoxia. These compounds have the potential to modify the TME, reducing extracellular acidification and potentially overcoming resistance to traditional treatments like chemotherapy and immunotherapy. Lymph node metastasis is a significant factor in the progression of OSCC, resulting from the cancer cell’s ability to migrate and invade local tissues [229]. PLAU, an enzyme involved in degrading the extracellular matrix to promote the invasion of tumor cells, has shown promise as a potential target for inhibitors such as amiloride (used for hypertension and heart failure), mexiletine (for treating ventricular arrhythmias), and 4-Chlorophenylguanidine (affecting ion channels) [230]. Understanding how these drugs affect immune cell infiltration and tumor immune evasion could lead to more effective therapeutic strategies. 

### 9.2. Overcoming Chemoresistance and Enhancing Immunotherapy Response 

Future research should aim to expand our understanding of how FDA-approved medications can be repurposed and combined with current therapies to improve OSCC outcomes. Investigating their effects on chemoresistance, immune modulation, and personalized treatment will be crucial in advancing OSCC therapies and ultimately improving patient survival and quality of life. These findings offer valuable insights into the precise management of OSCC. For instance, ANO1, identified from the cisplatin-resistant signature in the Lu et al. [165] study, could potentially be targeted by inhibitors such as fluoxetine, niflumic acid, and tannic acid. A recent study by Vyas et al. [231] unveiled how ANO1 can sequester cisplatin within lysosomes, preventing its access to the cell nucleus and thereby inducing chemoresistance in HNSCC. Tannic acid has been acknowledged as an inhibitor of ANO1 [232]. Darvin et al. [233] have demonstrated its anti-tumor properties explored in a preclinical model of OSCC. However, this study did not investigate the inhibition role of ANO1 in the OSCC preclinical model. As compared to niflumic acid and fluoxetine, the inhibitory effects of tannic acid on ANO1 have not yet been thoroughly investigated in terms of HNSCC cell migration inhibition in vitro [234]. Consequently, a combination of ANO1 inhibitors such as niflumic acid and fluoxetine with cisplatin may prove effective in overcoming chemo-resistance in preclinical models of HNSCC. Turning to immunotherapy resistance, we have pinpointed JTE-607 as an inhibitor of the immunosuppressive cytokine IL10. In a study by Chang et al. [235], the dual-action anti-CSF-1R-IL10 fusion protein, known as BF10, exhibited the ability to diminish tumor growth in a syngeneic preclinical model of HNSCC. Our identification of JTE-607 suggests it could serve as an alternative agent to curb tumor progression in vivo within the context of preclinical models of HNSCC, thereby enhancing the response to immune therapy. Overall, all FDA-approved drugs listed in Table 1, Table 2, Table 3, Table 4 and Table 5, along with their scientific names, commercial names, producers, years of approval, pharmaceutical formulations, doses, and mechanisms, are provided in Appendix A

## 10. Discussion

Overall, the data used for the analysis were sourced exclusively from online databases, underscoring the need for further in vivo and in vitro studies to corroborate these findings. This review acts as a crucial tool in enhancing the scientific comprehension of prognostic gene signatures in OSCC, especially through the integration of 34 studies utilizing LASSO Cox regression. By grouping gene signatures into major cancer pathways such as programmed cell death, epigenetic regulation, immune response, metabolism, and resistance to cell death, this study offers a structured framework for recognizing prognostic markers. The significance of conducting in vitro experiments to validate the potential risk or protective genes that are recognized as the prognostic markers involved in oral tumorigenesis has been underscored, as evidenced by a limited number of studies referenced by Chen et al. [183], Huang et al. [67], Gong et al. [75], Shi et al. [159], Wang et al. [169], and Yang et al. [88]. However, these studies have failed to confirm their in vitro findings utilizing animal models or explore the efficacy of potential inhibitors in targeting gene expression to enhance sensitivity to conventional treatment modalities like chemotherapy or immunotherapy. To bridge this gap in knowledge, further research utilizing in vitro methodologies, such as functional assays assessing cell proliferation through cell viability, colony formation, migration ability via wound healing assays, invasion potential through transwell assays, and immune–tumor cell co-culture systems to evaluate immunomodulatory effects, is crucial. In addition, employing patient-derived xenograft models for chemotherapy studies or syngeneic HNSCC mouse models for immunotherapy investigations is imperative to uncover the potential of inhibitors outlined in Table 1 in improving cisplatin sensitivity or modulating immune responses. These endeavors hold promise for augmenting the anti-tumor effects of ICIs and cisplatin therapy. This framework enriches the scientific knowledge base by combining disparate findings into a cohesive perspective, aiding in the discovery of new therapeutic targets. Furthermore, this review is of great importance for early-stage researchers, providing a detailed guide to understanding the intricate relationships between prognostic gene signatures and cancer pathways. It underscores particular methodological techniques, including LASSO Cox regression, which are essential for creating robust models. Additionally, by connecting these gene signatures to FDA-approved medications, the review closes the gap between biomarker discovery and clinical application, offering practical insights for translational research. This empowers emerging researchers not only to grasp the current landscape but also to pursue innovative avenues for future investigations in personalized cancer treatment.

## 11. Conclusions and Future Perspectives

In this review, we have comprehensively summarized and examined prognostic gene signatures in OSCC, organized into primary cancer pathways, including programmed cell death, immune response, metabolism, and resistance to apoptosis. These results highlight the promise of LASSO Cox regression-based models to enhance prognostic precision and guide treatment strategies tailored to the unique biological traits of OSCC. The findings presented in this review emphasize the significant role of immune-related and pathway-specific gene signatures in shaping the tumor microenvironment and affecting therapeutic outcomes. Given that OSCC poses a challenge in oncology due to its heterogeneity and mechanisms for immune evasion, these discoveries pave the way for creating predictive models and targeted therapies. Future investigations should concentrate on incorporating these gene signatures into personalized treatment regimens, utilizing FDA-approved medications when relevant, and validating these signatures in larger, more diverse patient populations. Furthermore, gaining insights into the dynamic interactions within the tumor microenvironment and their associations with clinical outcomes will further enhance prognostic models, ensuring their practical utility and effectiveness.

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
