# Peer review of "A Narrative Review of Prognostic Gene Signatures in Oral Squamous Cell Carcinoma Using LASSO Cox Regression"

_biomedicines, 2025, doi:10.3390/biomedicines13010134_

Round 1

Reviewer 1 Report

Comments and Suggestions for Authors

In the MS Biomedicines-3231697, the authors aimed to attract readers' interest through an impressive review of biomarkers implied in the survival prediction of OSCC patients. Moreover, they analyzed the FDA-approved medicines that use them as targets for OSCC therapy. From numerous studies, they selected only those that use LASSO-based Cox regression as a statistical tool. The MS is based on 235 references, over 50% of which were published in the last five years. The authors' merit is incontestable due to the high amount of information on the present MS.

The MS is structured as follows: 

1. Introduction (lines 41-203)

2. Prognostic Gene Signatures Related to Cancer Pathways in OSCC Using LASSO Cox Regression Analyses

2.1. Programmed Cell Death

2.2. Epigenetic and Gene Regulation

2.3. Metabolism and Energy

2.4. Resisting Cell Death

2.5. Immune Response and TME

3. The Prognostic Impact of Tregs in OSCC: Protective Role and Clinical Implications

4. Future Directions for FDA-Approved Drugs in OSCC

 5. Conclusion

The following comments and suggestions are available below:

I. As an overview, the MS contains many abbreviations. The authors are invited to list and explain them, which would be extremely helpful for the readers.

II. The impressive volume of data needs to be restructured. A review is a dynamic work that constantly maintains the reader's attention through various data presentation tools: tables, figures, and MS text organized in moderate-sized sections, subsections, and sub-subsections.

III. The data presented should be summarized at the beginning of each section to ensure a continuous interaction between authors and potential readers. 

IV. The MS title could be shortened, maintaining only the most relevant aspects; all removed data could be presented in the abstract, introduction, and discussion.

1. Introduction:

This section (lines 41-203) is hard to be deciphered in the current version. 

The authors are invited to organize the MS text in subsections with relevant titles.

2. Prognostic Gene Signatures Related to Cancer Pathways in OSCC Using LASSO Cox Regression Analyses

A brief presentation of statistical tools (considering pros and cons) used in survival prediction is requested to justify why the authors selected only the studies using LASSO Cox Regression. 

This section should be limited to the general data, and all subsections will become other individual sections. 

In the current version, Table 1 with mixed data is difficult to read; the reviewer has 2 suggestions:

- Classify the data displayed using a landscape page format.

Divide Table 1 into 3 smaller tables with suitably organized data and include them in each corresponding section; this could be a better option and more helpful for the reader.

3.  Programmed Cell Death (former 2.1.) could be Section 3 if the authors agree with the reviewer's suggestion, with a new Table 1.

According to the previous comment (III), a short paragraph is requested that presents a general overview of programmed cell death and all its types.

The authors should proceed similarly in each section: 

4. Epigenetic and Gene Regulation (Former 2.2.) should contain Table 2 and summary.

5. Metabolism and Energy (Former 2.3.) with Table 3 and a brief content presentation.

The authors should explain, in each table footer, the significance of the abbreviations used, 

6. Resting Cell Death (Former 2.4.) is substantially important. The authors are invited to make a short presentation of the significance of this phenomenon and all mechanisms implied in this process. Then, all aspects could be included as subsections.

7.  Immune Response and TME (former 2.5.) 

- please include the whole name of TME, not the abbreviation

Maybe this section should be divided into 3 subsections - please check and revise

Please find a more suitable title for "Immune" (line 1051).

8. The Prognostic Impact of Tregs in OSCC: Protective Role and Clinical Implications (former 3)

- It would be helpful to differentiate 2 subsections: first, with the protective role description, and second, with clinical applications, announced from the beginning. 

9. Future Directions for FDA-Approved Drugs in OSCC (former 4)

The authors are invited to present all FDA-approved drugs in a table with scientific names, commercial names, producers, years of approval, pharmaceutical formulation, dose, and mechanism.

This long section (1186-1290) could be divided into a few subsections. 

10. Discussion

The conclusions are too long in the present form. The Discussion section should include the first paragraph (1292-1300). Here, the authors should also provide evidence of how the present review could enrich the scientific database and help early researchers.  

11. Conclusions and future perspectives

The authors are invited to better synthesize the content from lines 1300-1315).

References:

Please edit the references in MDPI style.

Author Response

Reviewer 1

In the MS Biomedicines-3231697, the authors aimed to attract readers' interest through an impressive review of biomarkers implied in the survival prediction of OSCC patients. Moreover, they analyzed the FDA-approved medicines that use them as targets for OSCC therapy. From numerous studies, they selected only those that use LASSO-based Cox regression as a statistical tool. The MS is based on 235 references, over 50% of which were published in the last five years. The authors' merit is incontestable due to the high amount of information on the present MS.

The MS is structured as follows: 

  1. Introduction (lines 41-203)
  2. Prognostic Gene Signatures Related to Cancer Pathways in OSCC Using LASSO Cox Regression Analyses

2.1. Programmed Cell Death

2.2. Epigenetic and Gene Regulation

2.3. Metabolism and Energy

2.4. Resisting Cell Death

2.5. Immune Response and TME

  1. The Prognostic Impact of Tregs in OSCC: Protective Role and Clinical Implications
  2. Future Directions for FDA-Approved Drugs in OSCC
  3. Conclusion

The following comments and suggestions are available below:

  1. As an overview, the MS contains many abbreviations. The authors are invited to list and explain them, which would be extremely helpful for the readers.

Response 1: We have provided a list of abbreviations along with their full forms and associated functions relevant to this manuscript. This information is compiled in supplementary table 4, which includes the abbreviations and their corresponding explanations.

  1. The impressive volume of data needs to be restructured. A review is a dynamic work that constantly maintains the reader's attention through various data presentation tools: tables, figures, and MS text organized in moderate-sized sections, subsections, and sub-subsections.

Response 2: Thank you so much for your comments. We have taken your suggestion seriously and have made the amendment accordingly.

III. The data presented should be summarized at the beginning of each section to ensure a continuous interaction between authors and potential readers. 

Response 3: Thank you so much for your comments. We have taken your suggestion seriously and have made the amendment accordingly.

  1. The MS title could be shortened, maintaining only the most relevant aspects; all removed data could be presented in the abstract, introduction, and discussion.

Response 4: Thank you so much for your comments. We have shortened the title to “A Narrative Review of Prognostic Gene Signatures in Oral Squamous Cell Carcinoma Using LASSO Cox Regression”

  1. Introduction:

This section (lines 41-203) is hard to be deciphered in the current version. 

The authors are invited to organize the MS text in subsections with relevant titles.

Response 5: Thank you so much for your comments. We have included the subcategory for section 1.1 to 1.4.

  1. Prognostic Gene Signatures Related to Cancer Pathways in OSCC Using LASSO Cox Regression Analyses

A brief presentation of statistical tools (considering pros and cons) used in survival prediction is requested to justify why the authors selected only the studies using LASSO Cox Regression. 

Response 6: We have developed a table to simplify the brief presentation of statistical tools (considering pros and cons) used in survival prediction in supplementary table 1 (indicates at line 211) and to justify the reasons of selected only the studies using LASSO Cox Regression in supplementary table 2 (indicates at line 213).

This section should be limited to the general data, and all subsections will become other individual sections. 

Response 7: Thank you so much for the suggestions. We have made the amendment to according to your suggestion.

In the current version, Table 1 with mixed data is difficult to read; the reviewer has 2 suggestions:

- Classify the data displayed using a landscape page format.

Divide Table 1 into 3 smaller tables with suitably organized data and include them in each corresponding section; this could be a better option and more helpful for the reader.

Response 8: We agree with the reviewer’s suggestion. Accordingly, we have divided Table 1 into five smaller tables, each corresponding to Sections 3, 4, 5, 6, and 7, respectively. This reorganization ensures that the data is better classified and more reader friendly.

  1. Programmed Cell Death(former 2.1.) could be Section 3 if the authors agree with the reviewer's suggestion, with a new Table 1.

According to the previous comment (III), a short paragraph is requested that presents a general overview of programmed cell death and all its types.

The authors should proceed similarly in each section: 

Response 9: We agree with the reviewer’s suggestion and have added a short paragraph to present an overview of programmed cell death and all its types, which can be found on page 5, lines 233–250. Similarly, we have included an introductory paragraph for Sections 4 at line 529-552, 5 at line 722-743, and 6 at line 953-975 to ensure consistency and provide a comprehensive overview for the readers.

  1. Epigenetic and Gene Regulation(Former 2.2.) should contain Table 2 and summary.

Response 10: We have added table 2 for section 4 and we have included an introductory paragraph for Sections 4 at line 529-552.

  1. Metabolism and Energy(Former 2.3.) with Table 3 and a brief content presentation.

The authors should explain, in each table footer, the significance of the abbreviations used, 

Response 11: We have added table 3 for section 5 and we have included an introductory paragraph for section 5 at line 722-743. The significance of the abbreviations used can be found in supplementary table 4.

  1. Resting Cell Death (Former 2.4.) is substantially important. The authors are invited to make a short presentation of the significance of this phenomenon and all mechanisms implied in this process. Then, all aspects could be included as subsections.

Response 12: We have added table 4 for section 6 and we have included an introductory paragraph that describe the significance of this phenomenon and all mechanisms implied in this process at line 953-975.

  1. Immune Response and TME(former 2.5.) 

- please include the whole name of TME, not the abbreviation

Maybe this section should be divided into 3 subsections - please check and revise

Please find a more suitable title for "Immune" (line 1051).

Response 13: We have replaced the abbreviation "TME" with its full name at line 1140. Additionally, after careful consideration, we have divided this section into two subsections: 7.1 Immune Gene Signatures and Prognosis and 7.2 Arecoline-associated Fibrosis, instead of three subsections. Furthermore, we have updated the title of section 7.1 from "Immune" to Immune Gene Signatures and Prognosis for greater clarity and relevance.

  1. The Prognostic Impact of Tregs in OSCC: Protective Role and Clinical Implications(former 3)

- It would be helpful to differentiate 2 subsections: first, with the protective role description, and second, with clinical applications, announced from the beginning. 

Response 14: We thank the reviewer for the suggestions. We have divided Section 8 into two subsections: 8.1 Protective Role of Tregs in OSCC and 8.2 Clinical Applications of Tregs in OSCC Management.

  1. Future Directions for FDA-Approved Drugs in OSCC (former 4)

The authors are invited to present all FDA-approved drugs in a table with scientific names, commercial names, producers, years of approval, pharmaceutical formulation, dose, and mechanism.

This long section (1186-1290) could be divided into a few subsections. 

Response 15: We thank the reviewer for the suggestions. We have included all FDA-approved drugs with their scientific names, commercial names, producers, years of approval, pharmaceutical formulations, doses, and mechanisms in Supplementary Table 3, indicates at line 1432. Additionally, we have divided Section 9 into two subsections: 9.1 Targeting Hypoxia and Tumor Metabolism and 9.2 Overcoming Chemoresistance and Enhancing Immunotherapy Response.

  1. Discussion

The conclusions are too long in the present form. The Discussion section should include the first paragraph (1292-1300). Here, the authors should also provide evidence of how the present review could enrich the scientific database and help early researchers.  

Response 16: We thank the reviewer for the comments. We have revised the conclusion and made the necessary amendments accordingly. These amendments are reflected in Section 10, under the Discussion section.

  1. Conclusions and future perspectives

The authors are invited to better synthesize the content from lines 1300-1315).

Response 17: We thank the reviewer for the comments. We have revised the conclusion and made the necessary amendments accordingly. These amendments are reflected in Section 11, under the Conclusions and future perspectives section.

References:

Please edit the references in MDPI style.

Response 18: We have made the necessary amendments, and the references are now formatted according to the MDPI style.

Reviewer 2 Report

Comments and Suggestions for Authors

The article “A Systematic Review of Prognostic Gene Signatures in Oral Squamous Cell Carcinoma: Insights into Programmed Cell Death, Epigenetic Regulation, Immune Response, Metabolism, and Resistance to Cell Death Using LASSO Cox Regression” is very interesting and meticulously worked on.

The authors reflect in the title that the article is a systematic review, but methodologically it is not reproducible using the keywords they used for the selection of the studies.

The authors also do not follow the criteria (or protocol) of the systematic review guidelines, such as the one suggested by PRISMA (Preferred Reporting Items for Systematic reviews and Meta-Analyses/without meta-analyses, checklist), nor seem to have carried out the public registration, for example in PROSPERO.

It is essential to redirect the study methodology, enunciating the main objective(s) (or question) and reach the conclusions of each of the objectives.

Therefore, the topic can be reconsidered and a systematic review can be carried out using systematic review criteria or it can be approached as a narrative review that presents the perspective of new studies.

If the authors consider it appropriate, in line 718 they can add the reference, [47].

Understanding the extension of the present review, one of its limitations for a systematic review is that the search strategy was carried out in a single database.

Thank you very much

Author Response

Reviewer 2

The article “A Systematic Review of Prognostic Gene Signatures in Oral Squamous Cell Carcinoma: Insights into Programmed Cell Death, Epigenetic Regulation, Immune Response, Metabolism, and Resistance to Cell Death Using LASSO Cox Regression” is very interesting and meticulously worked on.The authors reflect in the title that the article is a systematic review, but methodologically it is not reproducible using the keywords they used for the selection of the studies.

The authors also do not follow the criteria (or protocol) of the systematic review guidelines, such as the one suggested by PRISMA (Preferred Reporting Items for Systematic reviews and Meta-Analyses/without meta-analyses, checklist), nor seem to have carried out the public registration, for example in PROSPERO.

It is essential to redirect the study methodology, enunciating the main objective(s) (or question) and reach the conclusions of each of the objectives.

Therefore, the topic can be reconsidered and a systematic review can be carried out using systematic review criteria or it can be approached as a narrative review that presents the perspective of new studies.

If the authors consider it appropriate, in line 718 they can add the reference, [47].

Understanding the extension of the present review, one of its limitations for a systematic review is that the search strategy was carried out in a single database.

Thank you very much

Response 1: We thank the reviewer for the insightful comments and suggestions. We agree that our approach does not follow the systematic review criteria. Therefore, we have revised the manuscript to reflect a narrative review rather than a systematic review.

We apologize for the confusion in the initial submission. To address this, we have adjusted the title and methodology of the manuscript to align with the narrative review approach. We have also clarified the main objectives of the review, reorganized the content to emphasize the narrative perspective, and ensured the conclusions are consistent with the revised methodology.

Additionally, we have acknowledged the limitation of conducting the search strategy in a single database and have explicitly stated this in the manuscript's limitations section. Furthermore, we have added the suggested reference [47] at line 718, as recommended, which is currently is known as reference [128] at line 793.

Thank you for helping us improve the clarity and accuracy of our manuscript.

Round 2

Reviewer 2 Report

Comments and Suggestions for Authors

Thank you very much for the corrections